# Pricing strategies for shared manufacturing platform considering cooperative advertising based on differential game

**Yantong Wu, Peng Liu** *

School of Management, Shenyang University of Technology, Shenyang, China

* liupeng@sut.edu.cn

## Abstract

Shared manufacturing is a new business form that focuses on all aspects of production and manufacturing, mainly relying on the shared manufacturing platform to achieve the optimal allocation of idle resources. For enterprises, in the process of deciding to lease idle resources, the pricing and advertising investment efficiency of the shared manufacturing platform is a valuable research issue. The shared manufacturing model in this paper consists of one manufacturer and one shared manufacturing platform, which will invest in cooperative advertising while the shared manufacturing process is completed. The cooperative advertising involves four models: the traditional cooperation model, the cost-sharing contract model, the revenue-sharing contract model, and the bilateral cost-sharing contract model. We investigate the impact of some key parameters on the prices and profits of the manufacturer and the shared manufacturing platform based on the differential game. The numerical examples demonstrate the viability of the model. Finally, we provide suggestions based on the decision-making of the manufacturer and the shared manufacturing platform under different cooperative advertising models.

## 1. Introduction

The development of shared manufacturing platforms has brought forth new trends for the expansion of the manufacturing industry. Reasonable pricing of shared manufacturing platform is the key to achieving a win-win shared manufacturing process. Advertising for products is becoming increasingly important as businesses strive to grow their market share and revenue. Companies and shared manufacturing platforms employ the effect of advertising to draw customers and improve product sales. A realistic cooperative advertising strategy is essential to reaching a win-win situation between the shared manufacturing platform and other players. Due in large part to the revolutionary spirit driving these platforms, more firms are now collaborating with some successful shared manufacturing platforms. Currently, the shared manufacturing platforms include four basic types: intermediary platform, collaborative platform, crowdsourcing platform and service-oriented platform. There are some shared platforms such as Predix of GE, Mindsphere of Siemens, ThingWorx of PTC, Tao Factory of Alibaba, and Hai Chuanghui of Haier.

**Data Availability Statement:** All relevant data are within the manuscript and its Supporting Information files.

**Funding:** This work was supported by the Research Project on Economic and Social

Development of Liaoning Province under Grant 2024lslybkt-058, Liaoning Graduate Education and Teaching Reform Research Project under Grant LNYJG2022062, the Key Program of Social Science Planning Foundation of Liaoning Province under Grant L21AGL017. The funders had no role in study design, data collection and analysis, decision to publish, or preparation of the manuscript.

**Competing interests:** The authors have declared that no competing interests exist.

**Fig 1. The operation mode of the intermediary shared manufacturing platform.**

A third-party platform that offers docking services between the supply and demand sides is the intermediary platform that is primarily examined in this study. The platform solely serves as a docking point and lacks production assets. The shared manufacturing platform gets profits by keeping the difference between its sales price and the producer's leasing fee per unit, while the manufacturer turns idle resources into income through the shared manufacturing platform. This type of intermediary shared manufacturing platform (as shown in Fig 1) does not own its own manufacturing resources and capabilities, such as workers, equipment, and materials. Manufacturing service providers transform idle manufacturing capacity to connect idle manufacturing capacity to the platform to form a virtual "cloud factory". The manufacturing buyer publishes the production order on the platform, and the supply and demand sides finally reach a deal through independent search and negotiation on the platform. The advantages of the intermediary sharing platform are that the platform is more flexible and the user entry threshold is low; The disadvantage is that Internet companies have a lack of understanding of the cross-border manufacturing industry. Typical application cases include "Haizhi Online", which is a leading non-standard parts manufacturing and sharing platform in China, which is mainly responsible for the docking of domestic small and medium-sized parts processing enterprises and global procurement resources. The buyer uploads the parts drawings and order information, calculates the corresponding average market price through the "intelligent price checker", and if the buyer accepts the price, it will match the appropriate factory for it, and finally the factory will make a specific offer. This model helps to shorten the procurement cycle and improve the production efficiency of the factory. In addition, there are Tao factories and Zhibu interconnection.

All parties are increasingly understanding the advantages of using promotional products to increase market share and revenue. The influence of advertising is used by manufacturers and shared manufacturing platforms to develop brand perception. For a business, investing in advertising is a crucial choice to increase the impact of corporate goodwill. It is essential to

utilize a fair advertising approach to achieve a win-win situation between a manufacturer and a shared production platform. The primary focus of advertising strategy study is cooperative advertising.

Since Ellen [1] proposed the concept of "shared manufacturing" for the development of small companies in 1990, it has attracted the attention of many scholars. Richard et al. [2] have extended the scope of shared manufacturing applications by applying shared manufacturing to the allocation and integration of large enterprise facility configurations. In recent years, many scholars have begun to study the basic concepts, functions, and classifications of shared manufacturing [3, 4]. Based on the definition and analysis of shared manufacturing, many scholars have combined different theories to establish different models of shared manufacturing. Yu et al. [5] used the BSM framework to integrate blockchain into shared manufacturing and demonstrated the feasibility of this combination. Rozman et al. [6] also integrated blockchain into shared manufacturing through a cross-chain solution. Zhang et al. [7] established an evolutionary game model of shared manufacturing with the participation of multiple subjects based on the theory of rank-dependent expected utility and evolutionary game. Zhang et al. [8] constructed a co-evolutionary game model of shared manufacturing quality synergistic improvement under the dynamic reward and punishment mechanism. Jiang and Li [9] used the shared factory as a production node to handle the concepts of shared orders, related resources, and production capacity. Li and Jiang [10] proposed a brand-new Enhanced Self-organizing Agent (ESA) in the context of shared communism. Wang et al. [11] established a digital twin-drive service model to solve the problems existing in the traditional shared manufacturing model, which performed better monitoring and control of shared manufacturing resources. Zhang et al. [12] proposed a service model that could monitor resources better. Ji et al. [13] started with parallel-machine scheduling and established a new model for the realization of shared manufacturing. Wei and Wu [14] studied two-machine hybrid flow-shop problems with fixed processing sequences based on the service arrangement of the shared manufacturing platform. Liu and Chen [15] considered the differential game model of shared manufacturing supply chain considering low-carbon emission reduction. Liu et al. [16] studied the optimal allocation of shared manufacturing resources based on bilevel programming. From the above literature, it can be seen that the difference between the discussion of pricing in the existing literature is that different state variables are considered, and different platform pricing models are established by changing different charging models or profit subjects. The essence of these models is to analyze the equilibrium solutions of the models under different cost sharing methods or benefit distribution methods, so as to provide a reference for supply chain or shared manufacturing enterprises. Therefore, this paper introduces cost-sharing contracts, revenue-sharing contracts, and bilateral cost-sharing contracts. This paper will deserve further research on its pricing strategy and related decisions.

In recent years, with the gradual maturity of network technology, research on pricing has begun to emerge. While most literature consider goodwill and emission reduction, many scholars consider the impact of state variables on platform pricing. You and Zhu [17] used product low-carbon and goodwill as state variables to model and use differential game to make pricing-related decisions. Zhu et al. [18] analyzed the pricing of cloud manufacturing platforms based on two-sided market theory. There are also many scholars who have classified and studied platform pricing from the perspective of different service models and channels. Li et al. [19] considered the pricing strategies for the shared manufacturing model based on the for-profit and not-for-profit cloud platforms. Hao et al. [20] established models under two operating model platforms to compare and analyze the decision-making results. Ye and Zhou [21] established a dual-channel supply chain model with product goodwill and emission reduction as state variables, and studied the decision-making behavior of each member in

different situations. Lu et al. [22] used the differential game to study dynamic pricing and technical innovation in the cloud service supply chain. Li et al. [23] studied the optimal pricing and coupon promotion policies under different channels based on the dual-channel supply chain. In the context of the cloud manufacturing supply chain, Yang et al. [24] used the differential game method to compare the optimal pricing and total profit in decentralized and centralized situations. Zhao and Chen [25] studied the pricing strategy of the monopoly capacity sharing platform in the context of cloud manufacturing and analyzed the profitability of the capacity sharing platform under the two modes of fixed per-transaction fee and registration fee. As can be seen from the above literature, the existing literature considers platform pricing decisions under different state variables, but does not share the closer partnerships that manufacturing platforms have emerged with manufacturers in reality. This paper will analyze the pricing decision problem under the advertising cooperation model between the manufacturer and the shared manufacturing platform under different cooperation contracts.

In addition, from the existing literature, the cooperative advertising is an important mean for manufacturers to expand market capacity and maximize self-interest. And collaborative advertising has been widely used in the supply chain. As early as 1976, scholars began to use differential games to construct cooperative advertising models, among which the Nerlove-Arrow goodwill model [26] and Sethi advertising model [27] were widely adopted. Based on this model, two models of non-cooperative advertising and cooperative advertising were established, and it was found that cooperative advertising could improve corporate profits and achieve Pareto improvement [28, 29]. The results of the study found that co-op advertising resulted in higher profits for manufacturers and retailers. According to the differential game model of cooperative advertising, it can also be found that cooperative advertising will not cause much damage to the brand image of retailer promotion, and can help manufacturers alleviate the competitive pressure of retailers' own brands [30, 31]. In most co-op advertising studies, the impact of advertising on market expansion and product sales is positive. Guo and Ma [32] applied the nonlinear dynamic system to a cooperative advertising model in supply chains and provided valuable insights into cooperative advertising programs. Simonov and Hill [33] measured the effectiveness of competitive advertising on brand keywords in sponsored search. Yao et al. [34] constructed a pricing decision model for the third-party collecting closed-loop supply chain with three advertising modes under the double factor hypothesis of advertising marketing efficiency factor and advertising cost factor. In recent years, scholars have begun to study the cooperative advertising model in the more complex supply chain. Huo and Wu [35] studied the competitive advertising strategy in the context of a dual-channel supply chain and obtained the impact of advertising investment. Chen and Zhang [36] studied the advertising investment of manufacturers and retailers under the dual-channel supply chain of brand competition, which consists of electronic direct sales channels and traditional retail channels. Wang et al. [37] studied the bilateral vertical cooperative advertising problem in a dual-channel supply chain by using Stackelberg game models. Kennedy et al. [38] established a three-level cooperative advertising supply chain model of manufacturer-distributor-retailer, and proved the advantages and feasibility of this supply chain. Chen et al. [39] used the dynamic programming equation method to obtain the optimal decision in the advertising model in different situations. The above literature mainly studies the cooperative advertising decisions under the cooperation and competition of different brands in the supply chain, and ignores the introduction of cooperative advertising pricing strategies under different contracts.

Based on the previous literature review, we have the following three points to summarize. Firstly, the attention of scholars has been drawn to the shared manufacturing problem and the idea of shared manufacturing has been widely accepted as a result of reviews and classifications

of the literature. Secondly, as shared manufacturing comes into people's awareness, the shared manufacturing platform pricing has gradually attracted the attention of researchers as a revenue party in the models. Thirdly, the existing literatures on the combination of dual-channel supply chain and cooperative advertising models have given a variety of models with different mathematical models. Through these summaries, we find that the investment in advertising directly affects the costs and benefits of enterprises in the supply chain. Through the cooperation between upstream and downstream, it can not only improve corporate profits, but also improve overall competitiveness. Therefore, for shared manufacturing platform enterprises, the input cost of advertising and the decision-making of different cooperative advertising methods are very worthy of study.

In summary, compared with the existing literature on shared manufacturing, the contributions of this paper are: (1) considering the long-term and dynamic characteristics of the impact of advertising investment on product demand, using differential games to study the pricing and advertising strategies under shared manufacturing from the perspective of long-term dynamics; (2) introducing a variety of contracts under the premise of cooperative advertising, and comparing and analyzing the decision-making of shared manufacturing enterprises under different contracts and the impact of different coefficients on the results.

## 2. Problem descriptions and assumptions

The manufacturer leases idle resources to the demander with the participation of the shared manufacturing platform, and both the manufacturer and the shared manufacturing platform will invest in advertising during the completion of the shared manufacturing process. In order to maximize profits, the traded products or resources must be more competitive. The two parties will choose a cooperation model for advertising investment. Under different models, the shared manufacturing platform and the manufacturer have corresponding decisions to maximize revenue. The model symbols and descriptions in this paper are as shown in Table 1.

This paper assumes that the shared manufacturing model under cooperative advertising consists of one manufacturer and one shared manufacturing platform. The products and idle resources that the manufacturer wants to rent will be entrusted to the shared manufacturing platform, and the shared manufacturing platform will perform secondary pricing based on manufacturer pricing to earn service fees by earning the difference. This is shown in Fig 2.

**Assumption 1.** Drawing on the Nerlove-Arrow model to describe the impact of manufacturers and retailers' advertising on market capacity, considering that the advertising

**Table 1. Model symbols and descriptions.**

| Symbol | Description |
|---|---|
| $t$ | Time ($t>0$) |
| $E_M(t)$, $E_C(t)$ | Advertising investment of manufacturer and shared manufacturing platform |
| $\omega$ | Manufacturer pricing ($0 \leq \omega \leq p$) |
| $p$ | Shared manufacturing platform pricing |
| $\mu_M, \mu_C$ | The sensitivity coefficients of market favorability to the advertising effort level of manufacturer and shared manufacturing platform, respectively. |
| $\delta$ | Natural decay rate ($\delta>0$) |
| $W(t)$ | The market capacity of unit product at time t |
| $D(p)$ | Requirements under different models |
| $C_M, C_C$ | Advertising input costs for the manufacturer and shared manufacturing platform |
| $\rho$ | Discount factor |
| $\theta_C, \theta_M, \varphi$ | Different contract model coefficients |

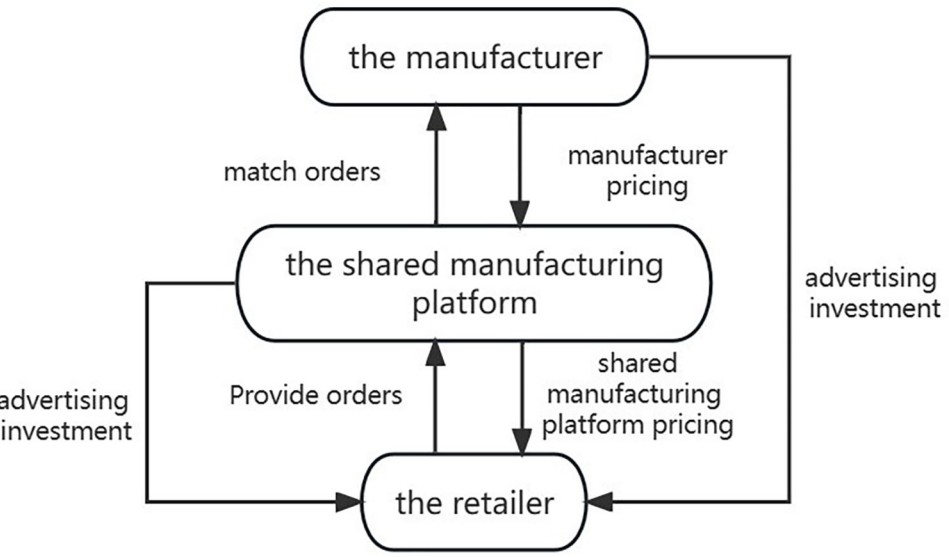

**Fig 2. Shared manufacturing system.**

investment of manufacturers and retailers will increase market capacity and ultimately affect consumer demand [40, 41], the change of market capacity satisfies the following differential equation:

$$\dot{W}(t) = \mu_M E_M(t) + \mu_C E_C(t) - \delta W(t) \tag{1}$$

where δ is the natural decay rate, which represents the impact of advertising on market capacity as it decays over time. And the sensitivity coefficients of market favorability to the advertising effort level of manufacturer and shared manufacturing platform are $\mu_M$ and $\mu_C$, respectively.

**Assumption 2.** Erickson [42] and Zhang et al [43] argue that both price and non-price factors have a linear relationship with market demand, while Ouardighi [44] argue that the above two factors have an impact on market demand through separation and multiplication. The price factor in this model composed of the manufacturer and the shared manufacturing platform is $h(p) = \alpha\text{-}\beta p$, so the demand function should be

$$D(W, p) = (\alpha - \beta p)\eta_W W(t) \tag{2}$$

where $\eta_W$, $\eta_G$ are the influence coefficients of market capacity brought by advertising investment on product demand, $D(W,p) \geq 0$, and since $h(p) \geq 0$, the price $p(t) \in [0, \frac{\alpha}{\beta}]$.

**Assumption 3.** Accounting the convexity of advertising costs [36], it is assumed that advertising cost is a convex function of advertising. Under the traditional cooperation model, the cost of the manufacturer and the shared manufacturing platform should be

$$\begin{cases} C_M^T(t) = \dfrac{L_M}{2}\left[E_M^T(t)\right]^2 \\[2mm] C_C^T(t) = \dfrac{L_C}{2}\left[E_C^T(t)\right]^2 \end{cases} \tag{3}$$

where $C_M^T(t)$, $C_C^T(t)$ are the cost of advertising effort of the manufacturer and the shared manufacturing platform at time $t$ and is a convex function of advertising investment; $L_M$, $L_C > 0$ are the cost coefficients of the advertising effort of the manufacturer and the shared

manufacturing platform. Similarly, it is assumed that the advertising investment does not change in unit product management costs. Therefore, the manufacturer's unit product production cost and management cost are regarded as constants, which are simplified to 0 for convenience.

**Assumption 4.** In the revenue-sharing contract model, part of the revenue of the shared manufacturing platform is shared to the manufacturer proportionally, and its sharing coefficient is denoted by $\varphi$, where $\varphi \in [0,1]$ ($\varphi = 0$ under other models, while $\varphi > 0$ under the revenue-sharing contract model).

**Assumption 5.** In different contract models (the traditional cooperation model, the cost-sharing contract model, and bilateral cost-sharing contract model), the cost-sharing coefficients of the manufacturer and the shared manufacturing platform are $\theta_M$, $\theta_C$, respectively.

1. In the traditional cooperation model, the manufacturer and the shared manufacturing platform do not subsidize the cost of advertising input to each other, namely, $\theta_M = 0$, $\theta_C = 0$.

2. In the cost-sharing contract model, the manufacturer shares the advertising input cost for the shared manufacturing platform, namely, $\theta_M^C > 0$, $\theta_C^C = 0$.

3. In the bilateral cost-sharing contract model, the manufacturer and the shared manufacturing platform share the cost of advertising input with each other, namely, $\theta_M^B > 0$, $\theta_C^B > 0$.

**Assumption 6.** It is assumed that the manufacturer and the shared manufacturing platform make rational decisions based on complete information, and ignore shared manufacturing's inventory costs and out-of-stock costs. The manufacturer and the shared manufacturing platform have the same discount factor $\rho$ in an infinite time horizon, where $\rho > 0$. Manufacturer pricing is denoted by a constant $\omega$, where $0 \le \omega \le p$.

The profits of the manufacturer and the shared manufacturing platform are as follows:

$$J_M = \max \int_0^\infty e^{-\rho t}[(\phi p + \omega)D(t) - C_M(t) + (1 - \theta_M)C_M(t)]dt$$
$$J_C = \max \int_0^\infty e^{-\rho t}[((1 - \phi)p - \omega)D(t) - C_C(t) + (1 - \theta_C)C_C(t)]dt$$

(4)

## 3. Model analysis and solution

### 3.1 Traditional cooperation model

Under the traditional cooperation model, the manufacturer and the shared manufacturing platform do not share costs with each other (as described in Hypothesis 4, the cost-sharing coefficient is 0). The aim is to determine the optimal level of advertising investment for manufacturers and shared manufacturing platforms to maximize their respective benefits. The "T" above indicates the optimal decision under traditional cooperation model.

Under the traditional cooperation decision, the profit function of the manufacturer and the shared manufacturing platform is:

$$J_M^T = \max_{E_M^T \ge 0} \int_0^\infty e^{-\rho t}[D^T(t) \cdot \omega - \frac{L_M}{2}(E_M^T)^2]dt$$

(5)

$$J_C^T = \max_{E_C^T \ge 0} \int_0^\infty e^{-\rho t}[D^T(t) \cdot (p - \omega) - \frac{L_C}{2}(E_C^T)^2]dt$$

(6)

**Theorem 1.** (1) Under the traditional cooperation model, the optimal strategies of the manufacturer and the shared manufacturing platform are as follows:

$$
\begin{cases}
p^{T*} = \dfrac{\alpha + \beta\omega}{2\beta} \\[2mm]
E_M^{T*} = \dfrac{\mu_M(\alpha - \beta\omega)\eta\omega}{2L_M(\rho + \delta)} \\[2mm]
E_C^{T*} = \dfrac{\mu_C(\alpha - \beta\omega)^2\eta}{4\beta L_C(\rho + \delta)}
\end{cases}
\tag{7}
$$

(2) The optimal trajectory of product market capacity is as follows:

$$
W^{T*}(t) = W_\infty^{T*} + (W_0 - W_\infty^{T*})e^{-\delta t}
\tag{8}
$$

$$
W_\infty^{T*} = \frac{\mu_M{}^2(\alpha - \beta\omega)\eta\omega}{2L_M(\rho + \delta)\delta} + \frac{\mu_C{}^2(\alpha - \beta\omega)^2\eta}{4\beta L_C(\rho + \delta)\delta}
$$

(3) Under the traditional cooperation model, the overall long-term profits of the manufacturer and the shared manufacturing platform are as follows:

$$
J_M^{T*} = e^{-\rho t}\left[\frac{\eta\omega(\alpha - \beta\omega)}{2(\rho + \delta)}W_0 + \frac{\eta^2\omega(\alpha - \beta\omega)^2}{8\rho(\rho + \delta)^2}\left(\frac{\mu_M{}^2\omega}{L_M} + \frac{\mu_C{}^2(\alpha - \beta\omega)}{\beta L_C}\right)\right]
\tag{9}
$$

$$
J_C^{T*} = e^{-\rho t}\left[\frac{\eta(\alpha - \beta\omega)^2}{4\beta(\rho + \delta)}W_0 + \frac{\eta^2(\alpha - \beta\omega)^3}{8\beta\rho(\rho + \delta)^2}\left(\frac{\mu_M{}^2\omega}{L_M} + \frac{\mu_C{}^2(\alpha - \beta\omega)}{4\beta L_C}\right)\right]
\tag{10}
$$

**Proof of Theorem 1.** According to the optimal control method, the optimal value functions of the long-term profit of the manufacturer and the shared manufacturing platform at time t are as follows:

$$
\begin{cases}
J_M^{T*}(W^T) = e^{-\rho t}V_M^T(W^T) \\[2mm]
J_C^{T*}(W^T) = e^{-\rho t}V_C^T(W^T)
\end{cases}
\tag{11}
$$

$V_M^T(W)$ and $V_C^T(W)$ satisfy the HJB equation for any $W \geqq 0$. In other words, it warrants maximizing the following equation:

$$
\rho V_M^T(W) = \max_{E_M^T \geq 0, E_C^T \geq 0}\left[(\alpha - \beta p^T)\eta W^T\omega - \frac{L_M}{2}(E_M^T)^2 + V_M^T(\mu_M E_M^T + \mu_C E_C^T - \delta W^T)\right]
\tag{12}
$$

$$
\rho V_C^T(W) = \max_{E_M^T \geq 0, E_C^T \geq 0}\left[(\alpha - \beta p^T)\eta W^T(p - \omega) - \frac{L_C}{2}(E_C^T)^2 + V_C^T(\mu_M E_M^T + \mu_C E_C^T - \delta W^T)\right]
\tag{13}
$$

To maximize $J_M^{T*}(W^T)$ and $J_C^{T*}(W^T)$, take the first-order partial derivative concerning $p^{T*}$, $E_M^{T*}$ and $E_C^{T*}$ on the right side of Eqs (12) and (13) and make it equal to 0.

$$\begin{cases} p^{T*} = \dfrac{\alpha + \beta\omega}{2\beta} \\[3mm] E_M^{T*} = \dfrac{\mu_M V_M^T}{L_M} \\[3mm] E_C^{T*} = \dfrac{\mu_C V_C^T}{L_C} \end{cases} \tag{14}$$

By substituting Eqs (14) into (12) and (13), we have

$$\rho V_M^{T*}(W^T) = \left[\frac{(\alpha - \beta\omega)\eta\omega}{2} - \delta V_M^T\right]W^T + \frac{\mu_M^2 V_M^{T\prime 2}}{2L_M} + \frac{\mu_C^2 V_M^T V_C^T}{L_C} \tag{15}$$

$$\rho V_C^{T*}(W^T) = \left[\frac{(\alpha - \beta\omega)^2\eta}{4\beta} - \delta V_C^T\right]W^T + \frac{\mu_M^2 V_M^T V_C^T}{L_M} + \frac{\mu_C^2 V_C^{T\prime 2}}{2L_C} \tag{16}$$

Looking at Eqs (15) and (16), it can be seen that the formula is a linear expression about $W^T$. So we can obtain:

$$\begin{cases} V_M^T = \dfrac{(\alpha - \beta\omega)\eta\omega}{2(\rho + \delta)} \\[3mm] V_C^T = \dfrac{(\alpha - \beta\omega)^2\eta}{4\beta(\rho + \delta)} \end{cases} \tag{17}$$

Substituting (17) into (14) can solve the solutions $E_M^{T*}$ and $E_C^{T*}$ of the manufacturer and the shared manufacturing platform under the traditional cooperation model, that is, (7). At this time, substituting $E_M^{T*}$ and $E_C^{T*}$ into (1) can solve the market capacity satisfies as $W^{T*}(t)$. Finally, substituting $E_M^{T*}$ and $E_C^{T*}$ into (12) and (13), the optimal profit function of the system as a whole under the traditional cooperation model (9) and (10) can be obtained.

Based on Theorem 1, $E_M$ and $E_C$ increase with the increase of market capacity coefficient $\eta$, meanwhile, natural decline rate $\delta$ and discount factor $\rho$, as well as cost coefficients $L_M$ and $L_C$, are all negatively correlated with corresponding advertising efforts.

**Corollary 1.** (1) The sensitivity coefficients of market favorability to the advertising effort level, $\mu_M$ and $\mu_C$, is directly proportional to the advertising efforts $E_M$ and $E_C$; (2) As manufacturer pricing, constant $\omega$, increases, the manufacturer's profit under the traditional cooperation model increases.

**Corollary 2.** The market capacity of the cooperation model is monotonic, and the price has no effect on the optimal trajectory of market capacity and its stable value.

Based on Formula (8), when $W_0 - W_\infty > 0$, the market capacity $W(t)$ decreases with the increase of time t; when $W_0 - W_\infty < 0$, the market capacity $W(t)$ increases with time t; when $W_0 - W_\infty = 0$, $W(t)$ is a constant.

## 3.2 Cost-sharing contract model

Based on the traditional cooperation between the manufacturer and the shared manufacturing platform advertising input, the cost-sharing contract model is that the manufacturer bears a certain coefficient and proportion of the advertising input cost of the shared manufacturing platform. The "C" above indicates the optimal decision under traditional cooperation model.

The costs and profits of the manufacturer and the shared manufacturing platform are as follows:

$$
\begin{cases}
C_M^C = \dfrac{L_M}{2}(E_M^C)^2 + \theta_M^C \dfrac{L_C}{2}(E_C^C)^2 \\[2mm]
C_C^C = (1 - \theta_M)\dfrac{L_C}{2}(E_C^C)^2
\end{cases}
\tag{18}
$$

$$
J_M^C = \max_{E_M^C \geq 0, E_C^C \geq 0} \int_0^\infty e^{-\rho t}[D^C(t) \cdot \omega - \frac{L_M}{2}(E_M^C)^2 - \theta_M^C \frac{L_C}{2}(E_C^C)^2]dt
\tag{19}
$$

$$
J_C^C = \max_{E_M^C \geq 0, E_C^C \geq 0} \int_0^\infty e^{-\rho t}[D^C(t) \cdot (p - \omega) - (1 - \theta_M^C)\frac{L_C}{2}E_C^{C2}(t)]dt
\tag{20}
$$

Solving the above decision model, Theorem 2 gives the result.

**Theorem 2.** (1) Under the cost-sharing contract model, the optimal strategies of the manufacturer and the shared manufacturing platform are as follows:

$$
\begin{cases}
p^{C*} = \dfrac{\alpha + \beta\omega}{2\beta} \\[3mm]
E_M^{C*} = \dfrac{\mu_M(\alpha - \beta\omega)\eta\omega}{2L_M(\rho + \delta)} \\[3mm]
E_C^{C*} = \dfrac{\mu_C(\alpha - \beta\omega)^2\eta}{4(1 - \theta_M^C)\beta L_C(\rho + \delta)}
\end{cases}
\tag{21}
$$

The optimal trajectory of product market capacity is as follows:

$$
W(t)^{C*} = W_\infty^{C*} + (W_0 - W_\infty^{C*})e^{-\delta t}
\tag{22}
$$

$$
W_\infty^{C*} = \frac{{\mu_M}^2(\alpha - \beta\omega)\eta\omega}{2\delta L_M(\rho + \delta)} + \frac{{\mu_C}^2(\alpha - \beta\omega)^2\eta}{4\delta(1 - \theta_M^C)\beta L_C(\rho + \delta)}
$$

(3) Under the cost-sharing contract model, the overall long-term profits of the manufacturer and the shared manufacturing platform are as follows:

$$
J_M^{C*} = e^{-\rho t}[\frac{\eta\omega(\alpha - \beta\omega)}{2(\rho + \delta)}W_0 + \frac{{\mu_M}^2\eta^2\omega^2(\alpha - \beta\omega)^2}{8L_M(\rho + \delta)^2} - \frac{\theta_M^C{\mu_C}^2(\alpha - \beta\omega)^4\eta^2}{32(1 - \theta_M^C)^2 L_C\beta^2(\rho + \delta)^2}
$$
$$
+ \frac{{\mu_C}^2(\alpha - \beta\omega)^3\omega\eta^2}{8(1 - \theta_M^C)L_C\beta(\rho + \delta)^2}]
\tag{23}
$$

$$
J_C^{C*} = e^{-\rho t}[\frac{\eta(\alpha - \beta\omega)^2}{4\beta(\rho + \delta)}W_0 + \frac{{\mu_C}^2(\alpha - \beta\omega)^4\eta^2}{32(1 - \theta_M^C)L_C\beta^2(\rho + \delta)^2} + \frac{{\mu_M}^2(\alpha - \beta\omega)^3\omega\eta^2}{8L_M\beta(\rho + \delta)^2}]
\tag{24}
$$

**Proof of Theorem 2.** According to the optimal control method, the optimal value functions of the long-term profit of the manufacturer and the shared manufacturing platform at time t are as follows:

$$
\begin{aligned}
J_M^{C*}(W) &= e^{-\rho t}V_M^C(W) \\
J_C^{C*}(W) &= e^{-\rho t}V_C^C(W)
\end{aligned}
\tag{25}
$$

$V_M^C(W)$ and $V_C^C(W)$ satisfy the HJB equation for any $W^C \geqq 0$. In other words, it warrants maximizing the following equation:

$$\rho V_M^C(W) = \max_{E_M^C \geq 0, E_C^C \geq 0} [(\alpha - \beta p^C)\eta W^C \omega - \frac{L_M}{2}(E_M^C)^2 - \theta_M^C \frac{L_C}{2}(E_C^C)^2 + V_M^C(\mu_M E_M^C + \mu_C E_C^C - \delta W^C)] \quad (26)$$

$$\rho V_C^C(W) = \max_{E_M^C \geq 0, E_C^C \geq 0} [(\alpha - \beta p^C)\eta W^C(p^C - \omega) - (1 - \theta_M^C)\frac{L_C}{2}E_C^{C2}(t) + V_C^C(\mu_M E_M^C + \mu_C E_C^C - \delta W^C)] \quad (27)$$

To maximize $J_M^{C*}(W^C)$ and $J_C^{C*}(W^C)$, take the first-order partial derivative concerning $p^{C*}$, $E_M^{C*}$ and $E_C^{C*}$ on the right side of Eqs (26) and (27) and make it equal to 0.

$$\begin{cases} p^{C*} = \dfrac{\alpha + \beta\omega}{2\beta} \\[3mm] E_M^{C*} = \dfrac{\mu_M V_M^C}{L_M} \\[3mm] E_C^{C*} = \dfrac{\mu_C V_C^C}{(1 - \theta_M^C)L_C} \end{cases} \quad (28)$$

By substituting Eqs (28) into (26) and (27), we have

$$\rho V_M^{C*}(W^C) = [\frac{(\alpha - \beta\omega)\omega\eta}{2} - \delta V_M^C]W^C + \frac{\mu_M^2 V_M^{C\prime 2}}{2L_M} - \frac{\mu_C^2 V_C^{C\prime 2}}{2(1 - \theta_M^C)^2 L_C} + \frac{\mu_C^2 V_M^c V_C^c}{(1 - \theta_M^C)L_C} \quad (29)$$

$$\rho V_C^{C*}(W^C) = [\frac{(\alpha - \beta\omega)^2\eta}{4\beta} - \delta V_C^C]W^C + \frac{\mu_C^2 V_C^{C\prime 2}}{2(1 - \theta_M^C)L_C} + \frac{\mu_M^2 V_M^C V_C^C}{L_M} \quad (30)$$

Looking at Eqs (29) and (30), it can be seen that the formula is a linear expression about $W^C$. So we can obtain:

$$\begin{cases} V_M^C = \dfrac{(\alpha - \beta\omega)\eta\omega}{2(\rho + \delta)} \\[3mm] V_C^C = \dfrac{(\alpha - \beta\omega)^2\eta}{4\beta(\rho + \delta)} \end{cases} \quad (31)$$

Substituting (31) into (28) can solve the solutions $E_M^{C*}$ and $E_C^{C*}$ of the manufacturer and the shared manufacturing platform under the cost-sharing contract model, that is, (21). At this time, substituting $E_M^{C*}$ and $E_C^{C*}$ into (1) can solve the market capacity satisfies as $W^{C*}(t)$. Finally, substituting $E_M^{C*}$ and $E_C^{C*}$ into (26) and (27), the optimal profit function of the system as a whole under the cost-sharing contract model (23) and (24) can be obtained.

## 3.3 Revenue-sharing contract model

The revenue-sharing contract model is based on the cooperation between the manufacturer and the shared manufacturing platform advertising investment. The shared manufacturing platform will share φ per unit of revenue to the manufacturer according to a certain percentage, which is derived from assumption 6. The costs and profits of the manufacturer and the

shared manufacturing platform are as follows:

$$
\begin{cases}
C_M^E = \dfrac{L_M}{2}(E_M^E)^2 \\[2mm]
C_C^E = \dfrac{L_C}{2}(E_C^E)^2
\end{cases}
\tag{32}
$$

$$
J_M^E = \max_{E_M^E \geq 0} \int_0^\infty e^{-\rho t}[D^E(t) \cdot (\phi p + \omega) - \frac{L_M}{2}(E_M^E)^2]dt
\tag{33}
$$

$$
J_C^E = \max_{E_C^E \geq 0} \int_0^\infty e^{-\rho t}[D^E(t) \cdot ((1-\phi)p - \omega) - \frac{L_C}{2}(E_C^E)^2]dt
\tag{34}
$$

**Theorem 3.** (1) Under the revenue-sharing contract model, the optimal strategies of the manufacturer and the shared manufacturing platform are as follows:

$$
\begin{cases}
p^{E*} = \dfrac{\alpha(1-\phi) + \beta\omega}{2\beta(1-\phi)} \\[3mm]
E_M^{E*} = \dfrac{\mu_M[(\alpha^2\phi + 2\alpha\beta\omega)(1-\phi)^2 - \beta^2\omega^2(2-\phi)]\eta}{4\beta L_M(1-\phi)^2(\rho+\delta)} \\[3mm]
E_C^{E*} = \dfrac{\mu_C[\alpha(1-\phi) - \beta\omega]^2\eta}{4(1-\phi)\beta L_C(\rho+\delta)}
\end{cases}
\tag{35}
$$

The optimal trajectory of product market capacity is as follows:

$$
W^{E*}(t) = W_\infty^{E*} + (W_0 - W_\infty^{E*})e^{-\delta t}
\tag{36}
$$

$$
W_\infty^{E*} = \frac{\mu_M^2[(\alpha^2\phi + 2\alpha\beta\omega)(1-\phi)^2 - \beta^2\omega^2(2-\phi)]\eta}{4\delta\beta L_M(1-\phi)^2(\rho+\delta)} + \frac{\mu_C^2[\alpha(1-\phi) - \beta\omega]^2\eta}{4\delta(1-\phi)\beta L_C(\rho+\delta)}
$$

(3) Under the revenue-sharing contract model, the overall long-term profits of the manufacturer and the shared manufacturing platform are as follows:

$$
\begin{aligned}
J_M^{E*} = e^{-\rho t}\Big[ & \frac{((\alpha^2\phi + 2\alpha\beta\omega)(1-\phi)^2 - \beta^2\omega^2(2-\phi))\eta}{4\beta(1-\phi)^2(\rho+\delta)}W_0 + \frac{\mu_M^2\eta^2((\alpha^2\phi + 2\alpha\beta\omega)(1-\phi)^2}{32L_M\beta^2(1-\phi)^4(\rho+\delta)^2} \\
& -\frac{\beta^2\omega^2(2-\phi))^2}{32L_M\beta^2(1-\phi)^4(\rho+\delta)^2} + \frac{\mu_C^2((\alpha^2\phi + 2\alpha\beta\omega)(1-\phi)^2 - \beta^2\omega^2(2-\phi))(\alpha(1-\phi) - \beta\omega)^2\eta^2}{16\beta^2(1-\phi)^3 L_C(\rho+\delta)^2}\Big]
\end{aligned}
\tag{37}
$$

$$
\begin{aligned}
J_C^{E*} = e^{-\rho t}\Big[ & \frac{(\alpha(1-\phi) - \beta\omega)^2\eta}{4\beta(1-\phi)(\rho+\delta)}W_0 + \frac{\mu_C^2\eta^2(\alpha(1-\phi) - \beta\omega)^4}{32L_C\beta^2(1-\phi)^2(\rho+\delta)^2} \\
& +\frac{\mu_M^2((\alpha^2\phi + 2\alpha\beta\omega)(1-\phi)^2 - \beta^2\omega^2(2-\phi))(\alpha(1-\phi) - \beta\omega)^2\eta^2}{16\beta^2(1-\phi)^3 L_M(\rho+\delta)^2}\Big]
\end{aligned}
\tag{38}
$$

**Proof of Theorem 3.** According to the optimal control method, the optimal value functions of the long-term profit of the manufacturer and the shared manufacturing platform at time t

are as follows:

$$\begin{cases} J_M^{E*}(W^E) = e^{-\rho t} V_M^E(W^E) \\ J_C^{E*}(W^E) = e^{-\rho t} V_C^E(W^E) \end{cases} \tag{39}$$

$V_M^E(W)$ and $V_C^E(W)$ satisfy the HJB equation for any $W^E \geqq 0$. In other words, it warrants maximizing the following equation:

$$\rho V_M^E(W) = \max_{E_M^E \geq 0, E_C^E \geq 0} [(\alpha - \beta p^E)\eta W^E(\phi p^E - \omega) - \frac{L_M}{2}(E_M^E)^2 + V_M^E(\mu_M E_M^E + \mu_C E_C^E - \delta W^E)] \tag{40}$$

$$\rho V_C^E(W) = \max_{E_M^E \geq 0, E_C^E \geq 0} [(\alpha - \beta p^E)\eta W^E((1-\phi)p^E - \omega) - \frac{L_C}{2}(E_C^E)^2 + V_C^E(\mu_M E_M^E + \mu_C E_C^E - \delta W^E)] \tag{41}$$

To maximize $J_M^{E*}(W^E)$ and $J_C^{E*}(W^E)$, take the first-order partial derivative concerning $p^{E*}$, $E_M^{E*}$ and $E_C^{E*}$ on the right side of Eqs (40) and (41) and make it equal to 0.

$$\begin{cases} p^{E*} = \dfrac{\alpha(1-\phi) + \beta\omega}{2\beta} \\ \\ E_M^{E*} = \dfrac{\mu_M V_M^E}{L_M} \\ \\ E_C^{E*} = \dfrac{\mu_C V_C^E}{L_C} \end{cases} \tag{42}$$

By substituting Eqs (42) into (40) and (41), we have

$$\rho V_M^{E*}(W^E) = [\frac{((\alpha^2\phi + 2\alpha\beta\omega)(1-\phi)^2 - \beta^2\omega^2(2-\phi))\eta}{4\beta(1-\phi)^2} - \delta V_M^E]W^E + \frac{\mu_M^2 V_M^{E\prime 2}}{2L_M} + \frac{\mu_C^2 V_M^E V_C^E}{L_C} \tag{43}$$

$$\rho V_C^{E*}(W^E) = [\frac{(\alpha(1-\phi) - \beta\omega)^2\eta}{4\beta(1-\phi)} - \delta V_C^E]W^E + \frac{\mu_M^2 V_M^E V_C^E}{L_M} + \frac{\mu_C^2 V_C^{E\prime 2}}{2L_C} \tag{44}$$

Looking at Eqs (43) and (44), it can be seen that the formula is a linear expression about $W^E$. So we can obtain:

$$\begin{cases} V_M^E = \dfrac{((\alpha^2\phi + 2\alpha\beta\omega)(1-\phi)^2 - \beta^2\omega^2(2-\phi))\eta}{4\beta(1-\phi)^2(\rho+\delta)} \\ \\ V_C^E = \dfrac{(\alpha(1-\phi) - \beta\omega)^2\eta}{4\beta(1-\phi)(\rho+\delta)} \end{cases} \tag{45}$$

Substituting (45) into (42) can solve the solutions $E_M^{E*}$ and $E_C^{E*}$ of the manufacturer and the shared manufacturing platform under the revenue-sharing contract model, that is, (35). At this time, substituting $E_M^{E*}$ and $E_C^{E*}$ into (1) can solve the market capacity satisfies as $W^{E*}(t)$. Finally, substituting $E_M^{E*}$ and $E_C^{E*}$ into (40) and (41), the optimal profit function of the system as a whole under the revenue-sharing contract model (37) and (38) can be obtained.

## 3.4 Bilateral cost-sharing contract model

The bilateral cost-sharing contract model is an extension of the cooperation model based on traditional cooperation. In this theory, not only does the manufacturer share the cost of the shared manufacturing platform, but the shared manufacturing platform also needs to share

the cost of the manufacturer. The manufacturer sharing coefficient is denoted by $\theta_M^B$, the shared manufacturing platform sharing coefficient is denoted by $\theta_C^B$. The sharing coefficient of the manufacturing platform is as shown in Assumption 5. The cost and profit functions of the two are shown as follows:

$$\begin{cases} C_M^B = (1 - \theta_C^B)\dfrac{L_M}{2}(E_M^B)^2 + \theta_M^B \dfrac{L_C}{2}(E_C^B)^2 \\ C_C^B = \theta_C^B \dfrac{L_M}{2}(E_M^B)^2 + (1 - \theta_M^B)\dfrac{L_C}{2}(E_C^B)^2 \end{cases} \tag{46}$$

$$J_M^B = \max_{E_M^B \geq 0, E_C^B \geq 0} \int_0^\infty e^{-\rho t}[D^B(t)\cdot\omega - (1 - \theta_C^B)\dfrac{L_M}{2}(E_M^B)^2 - \theta_M^B \dfrac{L_C}{2}(E_C^B)^2]dt \tag{47}$$

$$J_C^B = \max_{E_M^B \geq 0, E_C^B \geq 0} \int_0^\infty e^{-\rho t}[D^B(t)(p^B - \omega) - \theta_C^B \dfrac{L_M}{2}(E_M^B)^2 - (1 - \theta_M^B)\dfrac{L_C}{2}(E_C^B)^2]dt \tag{48}$$

**Theorem 4.** (1) Under the bilateral cost-sharing contract model, the optimal strategies of the manufacturer and the shared manufacturing platform are as follows:

$$\begin{cases} p^{B*} = \dfrac{\alpha + \beta\omega}{2\beta} \\ E_M^{B*} = \dfrac{\mu_M(\alpha - \beta\omega)\omega\eta}{2L_M(1 - \theta_M^B)(\rho + \delta)} \\ E_C^{B*} = \dfrac{\mu_C(\alpha - \beta\omega)^2\eta}{4(1 - \theta_C^B)\beta L_C(\rho + \delta)} \end{cases} \tag{49}$$

The optimal trajectory of product market capacity is as follows:

$$W^{B*}(t) = W_\infty^{B*} + (W_0 - W_\infty^{B*})e^{-\delta t} \tag{50}$$

$$W_\infty^{B*} = \dfrac{\mu_M{}^2(\alpha - \beta\omega)\omega\eta}{2\delta L_M(1 - \theta_M^B)(\rho + \delta)} + \dfrac{\mu_C{}^2(\alpha - \beta\omega)^2\eta}{4\delta(1 - \theta_C^B)\beta L_C(\rho + \delta)}$$

(3) Under the bilateral cost-sharing contract model, the overall long-term profits of the manufacturer and the shared manufacturing platform are as follows:

$$J_M^{B*} = e^{-\rho t}[\dfrac{\eta\omega(\alpha - \beta\omega)}{2(\rho + \delta)}W_0 + \dfrac{\mu_M{}^2\eta^2\omega^2(\alpha - \beta\omega)^2}{8(1 - \theta_C^B)L_M(\rho + \delta)^2} - \dfrac{\theta_M^B\mu_C{}^2(\alpha - \beta\omega)^4\eta^2}{32(1 - \theta_M^B)^2L_C\beta^2(\rho + \delta)^2}$$

$$+ \dfrac{\mu_C{}^2(\alpha - \beta\omega)^3\omega\eta^2}{8(1 - \theta_M^B)L_C\beta(\rho + \delta)^2}] \tag{51}$$

$$J_C^{B*} = e^{-\rho t}[\dfrac{\eta(\alpha - \beta\omega)^2}{4\beta(\rho + \delta)}W_0 + \dfrac{\mu_C{}^2(\alpha - \beta\omega)^4\eta^2}{32(1 - \theta_M^B)L_C\beta^2(\rho + \delta)^2} - \dfrac{\mu_M{}^2(\alpha - \beta\omega)^3\omega^2\eta^2}{8(1 - \theta_C^B)^2L_M(\rho + \delta)^2}$$

$$+ \dfrac{\mu_M{}^2(\alpha - \beta\omega)^3\omega\eta^2}{8(1 - \theta_C^B)L_M\beta(\rho + \delta)^2}] \tag{52}$$

**Proof of Theorem 4.** According to the optimal control method, the optimal value functions of the long-term profit of the manufacturer and the shared manufacturing platform at time t

are as follows:

$$\begin{cases} J_M^{B*}(W^T) = e^{-\rho t}V_M^B(W^B) \\ J_C^{B*}(W^T) = e^{-\rho t}V_C^B(W^B) \end{cases} \tag{53}$$

$V_M^B(W)$ and $V_C^B(W)$ satisfy the HJB equation for any $W^B \geqq 0$. In other words, it warrants maximizing the following equation:

$$\rho V_M^B(W) = \max_{E_M^B \geq 0, E_C^B \geq 0}[(\alpha - \beta p^B)\eta W^B\omega - (1 - \theta_C^B)\frac{L_M}{2}(E_M^B)^2 - \theta_M^B\frac{L_C}{2}(E_C^B)^2$$
$$+ V_M^B(\mu_M E_M^B + \mu_C E_C^B - \delta W^B)] \tag{54}$$

$$\rho V_C^B(W) = \max_{E_M^B \geq 0, E_C^B \geq 0}[(\alpha - \beta p^B)\eta W^B(p^B - \omega) - \theta_C^B\frac{L_M}{2}(E_M^B)^2 - (1 - \theta_M^B)\frac{L_C}{2}(E_C^B)^2$$
$$+ V_C^B(\mu_M E_M^B + \mu_C E_C^B - \delta W^B) \tag{55}$$

To maximize $J_M^{B*}(W^B)$ and $J_C^{B*}(W^B)$, take the first-order partial derivative concerning $p^{B*}$, $E_M^{B*}$ and $E_C^{B*}$ on the right side of Eqs (54) and (55) and make it equal to 0.

$$\begin{cases} p^{B*} = \dfrac{\alpha + \beta\omega}{2\beta} \\[2ex] E_M^{B*} = \dfrac{\mu_M V_M^B}{(1 - \theta_C^B)L_M} \\[2ex] E_C^{B*} = \dfrac{\mu_C V_C^B}{(1 - \theta_M^B)L_C} \end{cases} \tag{56}$$

By substituting Eq (56) into (54) and (55), we have

$$\rho V_M^{B*}(W^B) = [\frac{(\alpha - \beta\omega)\eta\omega}{2} - \delta V_M^B]W^B + \frac{{\mu_M}^2 V_M^{B'2}}{2(1 - \theta_C^B)L_M} - \frac{\theta_M^B {\mu_C}^2 V_C^{B'2}}{2(1 - \theta_M^B)L_C} + \frac{{\mu_C}^2 V_M^B V_C^B}{(1 - \theta_M^B)L_C} \tag{57}$$

$$\rho V_C^{B*}(W^B) = [\frac{(\alpha - \beta\omega)^2\eta}{4\beta} - \delta V_C^B]W^B + \frac{{\mu_C}^2 V_C^{B'2}}{2(1 - \theta_M^B)L_C} - \frac{\theta_C^B {\mu_M}^2 V_M^{B'2}}{2(1 - \theta_C^B)L_M} + \frac{{\mu_M}^2 V_M^B V_C^B}{(1 - \theta_C^B)L_M} \tag{58}$$

Looking at Eqs (57) and (58), it can be seen that the formula is a linear expression about $W^T$. So we can obtain:

$$\begin{cases} V_M^B = \dfrac{(\alpha - \beta\omega)\eta\omega}{2(\rho + \delta)} \\[2ex] V_C^B = \dfrac{(\alpha - \beta\omega)^2\eta}{4\beta(\rho + \delta)} \end{cases} \tag{59}$$

Substituting (59) into (56) can solve the solutions $E_M^{B*}$ and $E_C^{B*}$ of the manufacturer and the shared manufacturing platform under the bilateral cost-sharing contract model, that is, (49). At this time, substituting $E_M^{B*}$ and $E_C^{B*}$ into (1) can solve the market capacity satisfies as $W^{B*}(t)$. Finally, substituting $E_M^{B*}$ and $E_C^{B*}$ into (54) and (55), the optimal profit function of the system as a whole under the bilateral cost-sharing contract model (51) and (52) can be obtained.

**Table 2. Dynamic equation of market capacity of different models.**

| Model | Dynamic Equation | |
|---|---|---|
| Traditional cooperation | $W^{T*}(t) = W_\infty^{T*} + (W_0 - W_\infty^{T*})e^{-\delta t}$ | $W_\infty^T = \frac{\mu_M{}^2(\alpha-\beta\omega)\eta\omega}{2L_M(\rho+\delta)\delta} + \frac{\mu_C{}^2(\alpha-\beta\omega)^2\eta}{4\beta L_C(\rho+\delta)\delta}$ |
| Cost-sharing | $W^{C*}(t) = W_\infty^{C*} + (W_0 - W_\infty^{C*})e^{-\delta t}$ | $W_\infty^C = \frac{\mu_M{}^2(\alpha-\beta\omega)\eta\omega}{2\delta L_M(\rho+\delta)} + \frac{\mu_C{}^2(\alpha-\beta\omega)^2\eta}{4\delta(1-\theta_M^C)\beta L_C(\rho+\delta)}$ |
| Revenue-sharing | $W^{E*}(t) = W_\infty^{E*} + (W_0 - W_\infty^{E*})e^{-\delta t}$ | $W_\infty^E = \frac{\mu_M{}^2[(\alpha^2\phi + 2\alpha\beta\omega)(1-\phi)^2 - \beta^2\omega^2(2-\phi)]\eta}{4\delta\beta L_M(1-\phi)^2(\rho+\delta)}$ $+\frac{\mu_C{}^2[\alpha(1-\phi)-\beta\omega]^2\eta}{4\delta(1-\phi)\beta L_C(\rho+\delta)}$ |
| Bilateral cost-sharing | $W^{B*}(t) = W_\infty^{B*} + (W_0 - W_\infty^{B*})e^{-\delta t}$ | $W_\infty^B = \frac{\mu_M{}^2(\alpha-\beta\omega)\omega\eta}{2\delta L_M(1-\theta_C^B)(\rho+\delta)} + \frac{\mu_C{}^2(\alpha-\beta\omega)^2\eta}{4\delta(1-\theta_M^B)\beta L_C(\rho+\delta)}$ |

## 4. Comparative analysis

### 4.1 Comparative analysis of market capacity

The market capacity of each model is compared and analyzed. Table 2 shows the dynamic equation of market capacity of different models obtained in the previous part.

**Corollary 3.** Since $\theta$ and $\varphi \in (0,1)$, then $(1-\theta) \in (0,1)$ and $(1-\varphi) \in (0,1)$. If the contract coefficients are the same, they can be compared, namely, $W_\infty^E > W_\infty^B > W_\infty^C > W_\infty^T$.

### Proof of Corollary 3

Taking bilateral cost-sharing and cost-sharing decision-making as an example,

$$W_\infty^B - W_\infty^C = \frac{\mu_M{}^2(\alpha-\beta\omega)\omega\eta}{2\delta L_M(1-\theta_C^B)(\rho+\delta)} + \frac{\mu_C{}^2(\alpha-\beta\omega)^2\eta}{4\delta(1-\theta_M^B)\beta L_C(\rho+\delta)} - \frac{\mu_M{}^2(\alpha-\beta\omega)\eta\omega}{2\delta L_M(\rho+\delta)} +$$
$$\frac{\mu_C{}^2(\alpha-\beta\omega)^2\eta}{4\delta(1-\theta_M^C)\beta L_C(\rho+\delta)} = \frac{\mu_M{}^2(\alpha-\beta\omega)\omega\eta}{2\delta L_M(\rho+\delta)} \cdot \frac{1}{(1-\theta_C)} + \frac{\mu_C{}^2(\alpha-\beta\omega)^2\eta}{4\delta(1-\theta_M)\beta L_C(\rho+\delta)}$$ Because
$(1-\theta_m), (1-\theta_C) > 0$, $W_\infty^B - W_\infty^C > 0$.

Similarly, $W_\infty^C - W_\infty^T > 0$, $W_\infty^E - W_\infty^B > 0$.

Given the initial value of market capacity, (1) when $W_0$ is constant, if and only if $W_0 > W_\infty$, the trajectory of market capacity increases over time, showing a downward trend. (2) Given $W_0$, if and only if $W_0 < W_\infty$, the market capacity trajectory increases with time, showing an upward trend. When the time increases to a certain extent, the market capacity converges to a stable point, namely $W_\infty$.

### 4.2 Analysis of price p

We analyze the factors that affect the price and then analyze the price of each model to obtain theorem 4. Table 3 shows the price decisions of different models obtained in the previous part, and Table 4 shows the results of the parameter impact analysis.

**Table 3. Price decisions of different models.**

| Model | Price |
|---|---|
| Traditional cooperation | $p^{T*} = \frac{\alpha+\beta\omega}{2\beta}$ |
| Cost-sharing | $p^{C*} = \frac{\alpha+\beta\omega}{2\beta}$ |
| Revenue-sharing | $p^{E*} = \frac{\alpha(1-\phi)+\beta\omega}{2\beta(1-\phi)}$ |
| Bilateral cost-sharing | $p^{B*} = \frac{\alpha+\beta\omega}{2\beta}$ |

**Table 4. The price changes with the coefficient.**

| Model | α increase | β increase | ω increase | φ increase |
|---|---|---|---|---|
| Traditional cooperation $p^{T*}$ | increases | decreases | increases | no effect |
| Cost-sharing $p^{C*}$ | increases | decreases | increases | no effect |
| Revenue-sharing $p^{E*}$ | increases | decreases | increases | decreases |
| Bilateral cost-sharing $p^{B*}$ | increases | decreases | increases | no effect |

### Corollary 4

(1) $p^{E*} > p^{T*} = p^{C*} = p^{B*}$.

(2) The price increases with the increase of α and ω, and decreases with the increase of β and φ.

### Proof of Corollary 4

$$p^{E*} - p^{T*} = \frac{\alpha(1-\phi) + \beta\omega}{2\beta(1-\phi)} - \frac{\alpha + \beta\omega}{2\beta} = \frac{\omega\phi}{2(1-\phi)} > 0.$$

This is because the other three models are subsidized in advertising costs, and the revenue sharing model is to share the difference in price obtained by the shared manufacturing platform with the manufacturer proportionally, which directly affects the pricing of the shared manufacturing platform. At the same time, the shared manufacturing platform will increase the pricing on the shared manufacturing platform to make up for the profits shared with the manufacturer.

### 4.3 Manufacturer's pricing ω and decision

The following conclusions can be drawn from Table 5:

### Corollary 5

(1) In the four models, the manufacturer's profit increases with the increase of the influence coefficient $\eta$ of market capacity on product demand;

**Table 5. Manufacturer's profit.**

| Model | manufacturer's profit |
|---|---|
| Traditional cooperation | $J_M^{T*} = e^{-\rho t}\left[\frac{\eta\omega(\alpha-\beta\omega)}{2(\rho+\delta)}W_0 + \frac{\eta^2\omega(\alpha-\beta\omega)^2}{8\rho(\rho+\delta)^2}\left(\frac{\mu_M{}^2\omega}{L_M} + \frac{\mu_C{}^2(\alpha-\beta\omega)}{\beta L_C}\right)\right]$ |
| Cost-sharing | $J_M^{C*} = e^{-\rho t}\left[\frac{\eta\omega(\alpha-\beta\omega)}{2(\rho+\delta)}W_0 + \frac{\mu_M{}^2\eta^2\omega^2(\alpha-\beta\omega)^2}{8L_M(\rho+\delta)^2} - \frac{\theta_M^C\mu_C{}^2(\alpha-\beta\omega)^4\eta^2}{32(1-\theta_M^C)^2L_C\beta^2(\rho+\delta)^2}\right.$ $\left. + \frac{\mu_C{}^2(\alpha-\beta\omega)^3\omega\eta^2}{8(1-\theta_M^C)L_C\beta(\rho+\delta)^2}\right]$ |
| Revenue-sharing | $J_M^{E*} = e^{-\rho t}\left[\frac{[(\alpha^2\phi+2\alpha\beta\omega)(1-\phi)^2 - \beta^2\omega^2(2-\phi)]\eta}{4\beta(1-\phi)^2(\rho+\delta)}W_0 + \frac{\mu_M{}^2\eta^2[(\alpha^2\phi+2\alpha\beta\omega)(1-\phi)^2}{32L_M\beta^2(1-\phi)^4(\rho+\delta)^2}\right.$ $\left. - \frac{\beta^2\omega^2(2-\phi)]^2}{32L_M\beta^2(1-\phi)^4(\rho+\delta)^2} + \frac{\mu_C{}^2[(\alpha^2\phi+2\alpha\beta\omega)(1-\phi)^2 - \beta^2\omega^2(2-\phi)][\alpha(1-\phi)-\beta\omega]^2\eta^2}{16\beta^2(1-\phi)^3L_C(\rho+\delta)^2}\right]$ |
| Bilateral cost-sharing | $J_M^{B*} = e^{-\rho t}\left[\frac{\eta\omega(\alpha-\beta\omega)}{2(\rho+\delta)}W_0 + \frac{\mu_M{}^2\eta^2\omega^2(\alpha-\beta\omega)^2}{8(1-\theta_C^B)L_M(\rho+\delta)^2} - \frac{\theta_M^B\mu_C{}^2(\alpha-\beta\omega)^4\eta^2}{32(1-\theta_M^B)^2L_C\beta^2(\rho+\delta)^2}\right.$ $\left. + \frac{\mu_C{}^2(\alpha-\beta\omega)^3\omega\eta^2}{8(1-\theta_M^B)L_C\beta(\rho+\delta)^2}\right]$ |

(2) With the increase of the manufacturer's price ω, $J_M^T$, $J_M^C$, $J_M^B$ continue to increase and $J_M^E$ continue to decrease.

### Proof of Corollary 5

Taking traditional cooperation decision-making as an example, for the sake of calculation, without affecting the practical significance, let's assume:

$$\mu_M = \mu_C = \mu, \ L_M = L_C = L.$$

At this point

$$\frac{\partial J_M^{T*}}{\partial \omega} = A \cdot (\alpha - 2\beta\omega) + B \cdot (\alpha - \beta\omega)(\alpha - 3\beta\omega)$$

$$= 3\beta^2 \cdot B \cdot \omega^2 - (2 \cdot A + 4\alpha \cdot B)\beta \cdot \omega + 2 \cdot A + \alpha^2 \cdot B$$

$$A = \frac{e^{-\rho t}\eta W_0}{2(\rho + \delta)}, B = \frac{e^{-\rho t}\alpha\eta^2\mu^2}{8L\beta\rho(\rho + \delta)^2}$$

According to the function theorem, we can analyze that $J_M^{T*}$ is a quadratic function with respect to ω. And then according to Vieda's theorem and $\omega \in (0, \frac{z}{\beta})$, we can find out:

When $0<\omega<\tau$, $J_M^{T*} > 0$; when $\tau < \omega < p < \frac{\alpha}{\beta}$, $J_M^{T*} < 0$.

where: $\tau = \frac{(2 \cdot A + 4\alpha \cdot B)\beta + \sqrt{(2 \cdot A + 4\alpha \cdot B)^2\beta^2 - 12\beta^2(\alpha \cdot A + \alpha^2 \cdot B)}}{6\beta^2 \cdot B}$ therefore, when $0<\omega<\tau$, $J_M^{T*}$ increases; when $\tau < \omega < p < \frac{\alpha}{\beta}$, $J_M^{T*}$ decreases.

That is, with the increase of the manufacturer's price ω, the profit of the manufacturer increases first and then decreases.

The four models' maximum profits and growth rates differ due to the influence of the advertising cooperation contract coefficient, which causes the comparison of the manufacturer's early-stage profits to provide the following results: $J_M^E > J_M^T > J_M^B > J_M^C$. When the manufacturer's pricing ω rises, the manufacturer's profit comparison at this time is $J_M^B > J_M^C > J_M^T > J_M^E$, so the manufacturer should select the revenue-sharing contract model when price ω is low and select the bilateral cost-sharing contract model when pricing ω is high.

## 5. Numerical analysis

In order to verify the above Corollaries and the equilibrium solution results under different models, this section analyzes the relevant conclusions of idle capacity providers and shared manufacturing platforms in four game cases by assigning exogenous variables to verify the effectiveness of the model. This section will conduct numerical analysis to more intuitively compare the profits of the manufacturer and shared manufacturing platform under different models and decisions, as well as the impact of key parameters on pricing and profits. Based on the literature [25, 33], we set the relevant parameters as $\alpha = 5$, $\beta = 2$, $\eta = 0.75$, $\delta = 0.2$, $\mu_M = \mu_C = 0.5$, $L_M = L_C = 1$, $\rho = 0.3$, $\omega = 2$, $\theta = \varphi = 0.4$, $\gamma_M = \gamma_C = 0.5$, $K_M = K_C = 1$, $W_0 = 10$, $G_0 = 15$.

### 5.1 Comparison of dynamic equations

The dynamic equations in the four cases are shown in Fig 3. Over time, the market capacity $W$ $(t)$ under whichever model decreases with the increase of time $t$, and finally reaches a steady state. This is because $W_\infty < W_0$ exists in these four models, so the monotonicity of $W(t)$ decreases with time increases, which verifies Corollary 2.

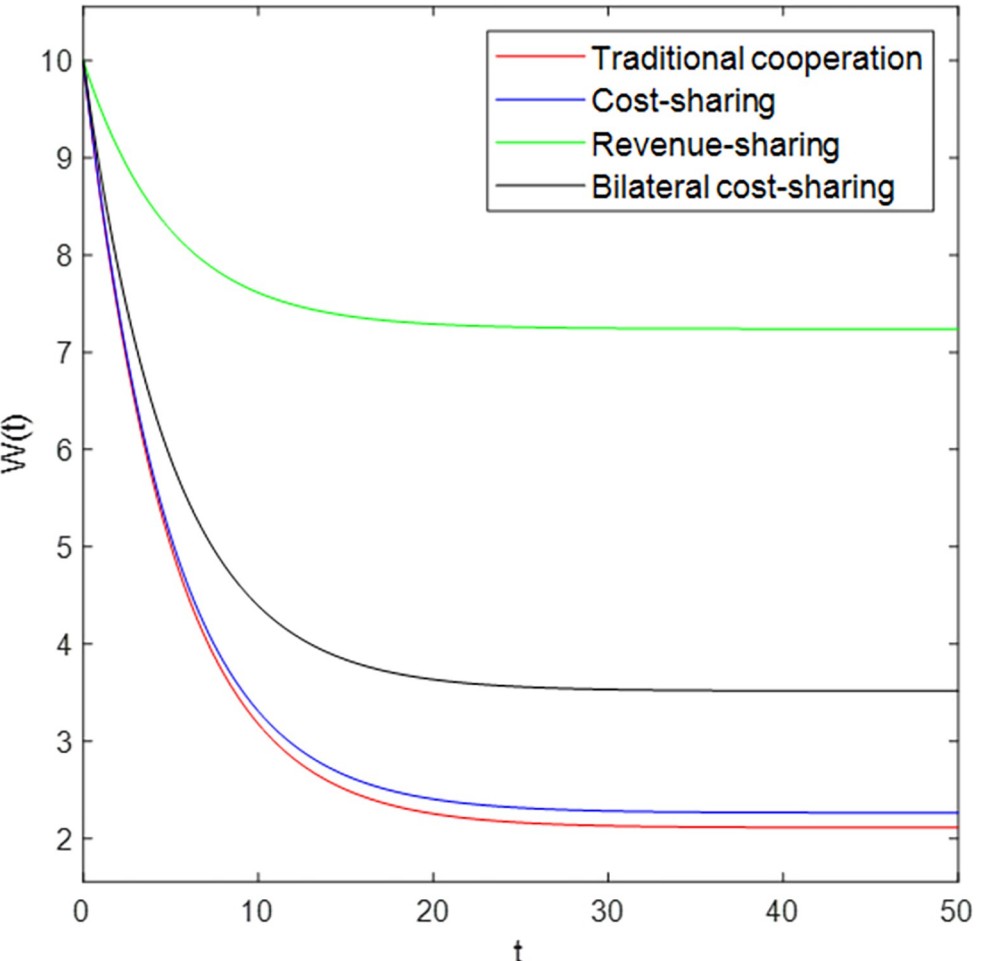

**Fig 3. Changes in market capacity trajectories over time t.**

Fig 3 shows that among the four different models, the revenue-sharing model maintains a higher market capacity in a steady-state, followed by the bilateral cost-sharing model, the cost-sharing model, and the traditional cooperation model. That is, under the traditional cooperation model, when the time increases to a certain extent because the manufacturer has the lowest cooperation relationship with the shared manufacturing platform, the market capacity it maintains is also the lowest, which verifies Corollary 3.

### 5.2 Influence of key parameters on p

The relationship between key parameters and price is shown in Figs 4–6. As the abscissa $\alpha$, $\omega$, and $\varphi$ increase, the ordinate price $p$ also increases, which is positively correlated. Among them, $\omega$ has the greatest influence on the price. Figs 4–6 show that with the increase of $\beta$, the price $p$ decreases and becomes negatively correlated, verifying Corollary 4.

### 5.3 Profit comparison under different models

Taking the traditional cooperation model and revenue-sharing contract model as examples (the cost-sharing model and bilateral cost-sharing model have similar trajectories to the traditional cooperation model), it can be concluded in Figs 7 and 8 that the shared manufacturing

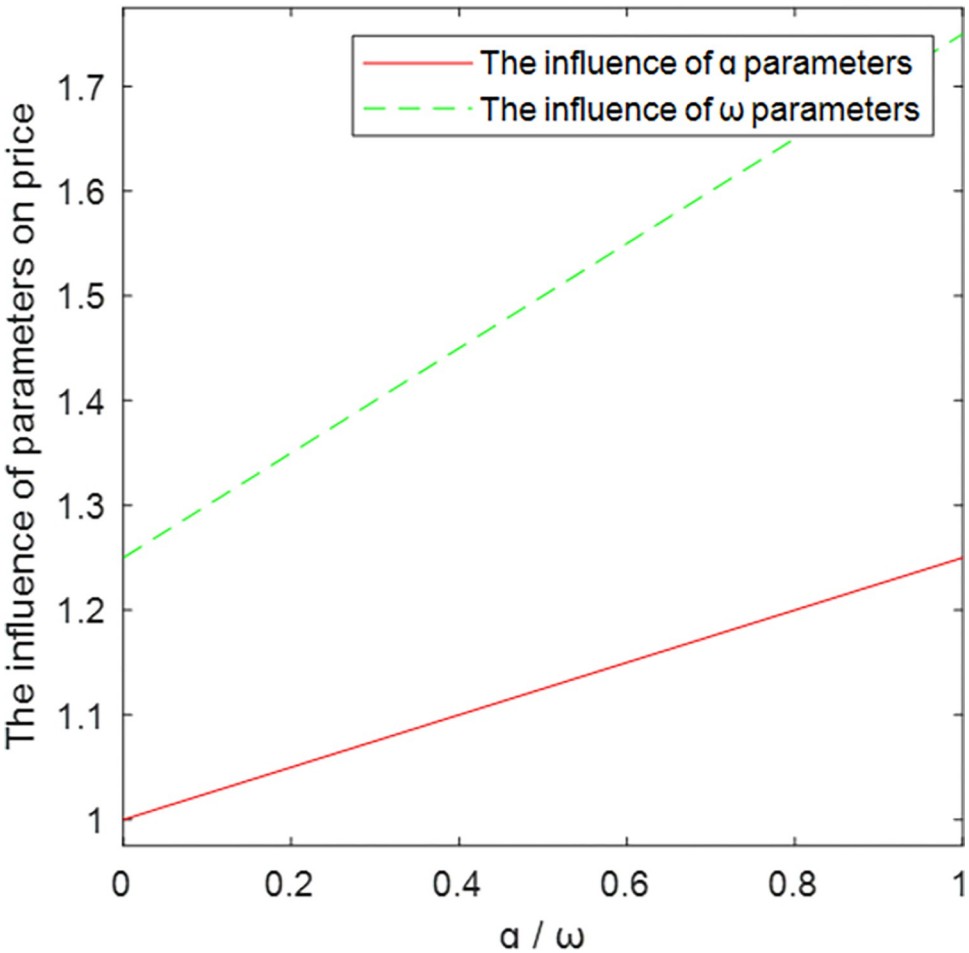

**Fig 4. The influence of $\alpha$ and $\omega$ parameters on price.**

platform's profit always decreases with the increase of $\omega$, while manufacturer shows a trend of increasing first and then decreasing, verifying Corollary 1 and Corollary 5. The results show that the price difference between manufacturers and shared manufacturing platforms is larger. That is, the higher the price of idle resources on the shared manufacturing platform, and the shared manufacturing platform's profit is decreasing; for the manufacturer, when the increase reaches a certain level with the increase of $\omega$, due to the increasingly high pricing of the shared manufacturing platform, the rented resources will no longer have market advantages, resulting in the manufacturer's profits beginning to dwindle.

### 5.4 Manufacturer's pricing ω and decision

Fig 9 can be seen that the comparison result of the profit obtained by different models of the shared manufacturing platform is to verify Corollary 5.

Fig 10 shows that when the manufacturer's price ω takes different values, the comparison results of the profits obtained are different. When $\omega \in [0,0.69]$, $J_M^E > J_M^T > J_M^B > J_M^C$; when $\omega \in [0.69,23.05]$, $J_M^E > J_M^B > J_M^T > J_M^C$; when $\omega \in [23.05,1.03]$, $J_M^E > J_M^B > J_M^C > J_M^T$. When the manufacturer price $\omega \in [0, 1.03]$, the manufacturer chooses the revenue sharing model; and $\omega \in [1.03, 1.2]$, the manufacturer chooses the bilateral cost-sharing model.

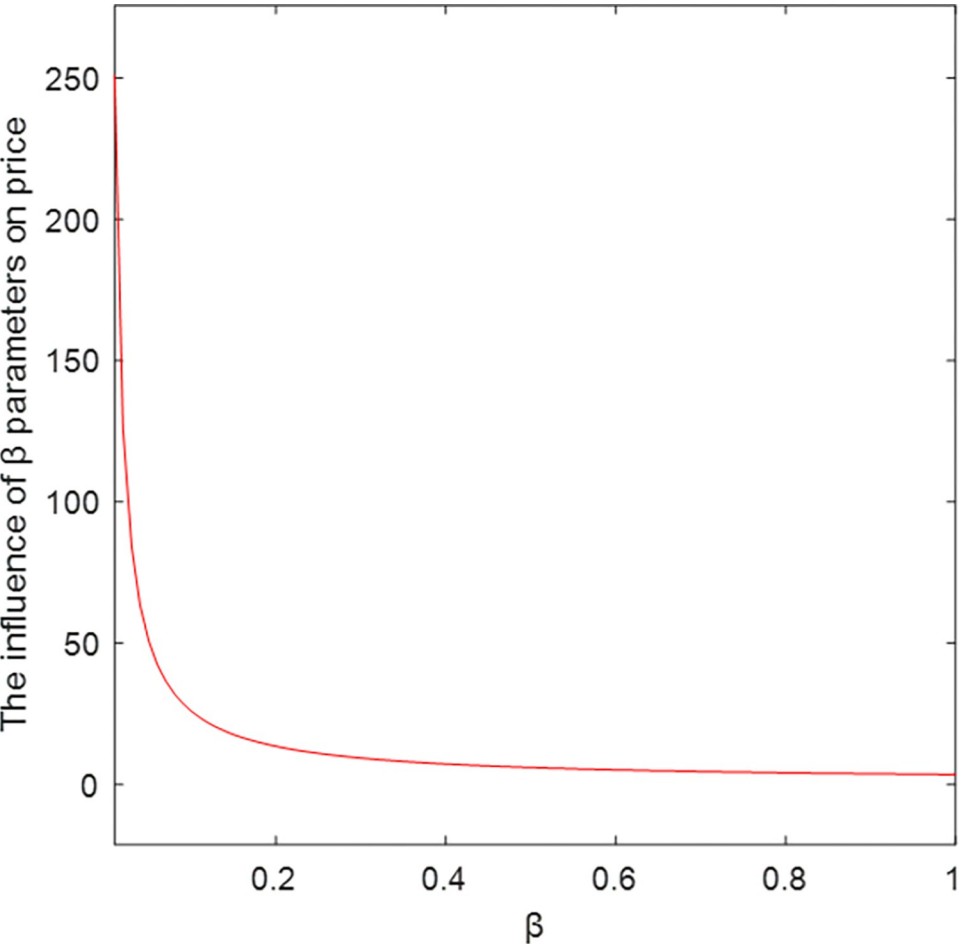

**Fig 5. The influence of β parameters on price.**

As can be seen from this section, the numerical simulation results support the inferences described in the previous chapters, and successfully verify the validity of the model established in this paper.

## 6. Conclusions

This paper analyzes and compares the pricing strategies and the different decision-making behaviors of the manufacturer and the shared manufacturing platform considering the traditional cooperation model, the cost-sharing contract model, the revenue-sharing contract model and the bilateral cost-sharing contract model.

Based on the theory of cooperative advertising, this paper introduces the sensitivity coefficient of the market to the advertising efforts of manufacturers and shared manufacturing platforms, and studies the optimal pricing strategy of shared manufacturing platforms. By establishing the traditional cooperation model, the cost-sharing contract model, the revenue-sharing contract model and the bilateral cost-sharing contract model, we discuss the influence of manufacturers' pricing, market sensitivity coefficient to advertising efforts and time on the equilibrium profit and optimal pricing decision of the shared manufacturing platform. And specific conclusions are given: (1) As time increases when the trajectory of the dynamic equation of market capacity converges, the market capacity maintained by different models from

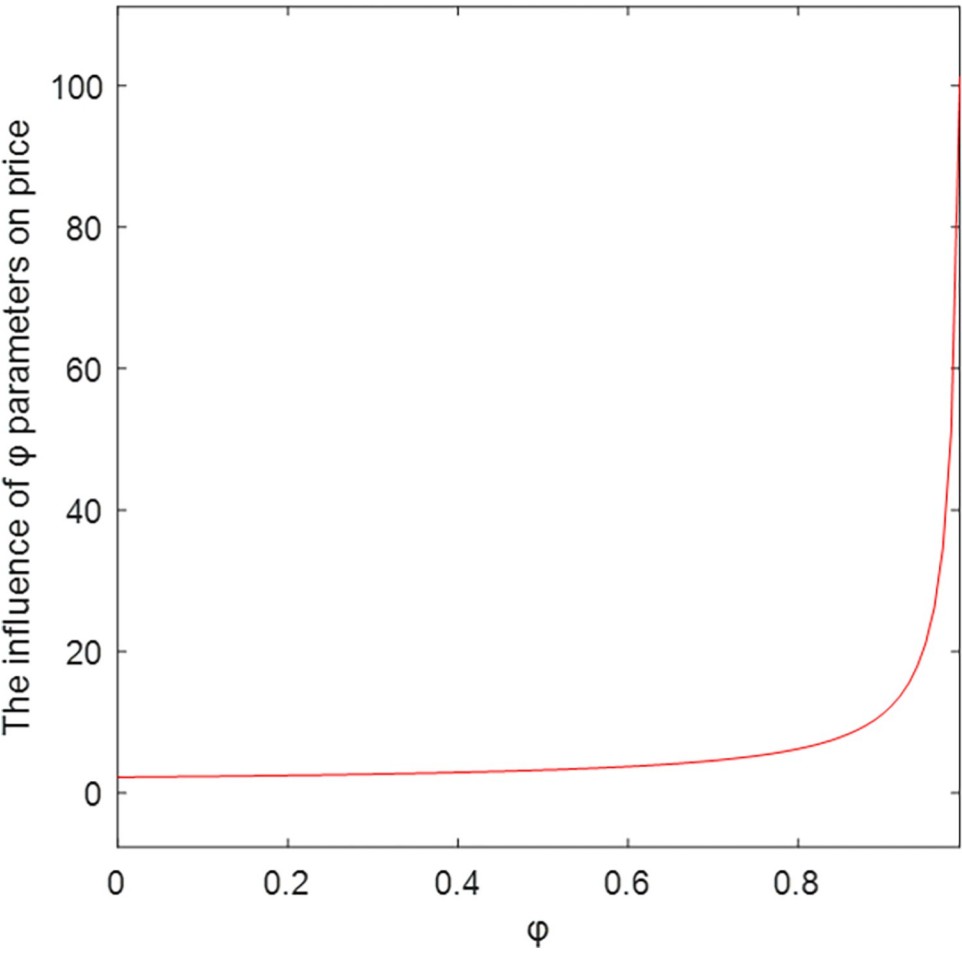

**Fig 6. The influence of φ parameters on price.**

high to low is still the revenue-sharing contract model, the bilateral cost-sharing contract model, the cost-sharing contract model and the traditional cooperation model. It is explained that the cooperation effect of advertising investment can be improved by introducing contracts, and the effect of different contracts is different, among which the revenue sharing contract can maximize the market capacity of the product; (2) When other coefficients are the same as the manufacturer's pricing, the shared manufacturing platform pricing p of the traditional cooperation model, the cost-sharing contract model and the bilateral cost-sharing contract model remains unchanged, while the shared manufacturing platform pricing is higher under the revenue-sharing model; (3) An increase in manufacturer pricing leads to a first increase followed by a decrease in the profit of the manufacturer, while the profit of the shared manufacturing platform is decreasing. It is illustrated that the lower the manufacturer prices, the higher the shared manufacturing revenue.

The research findings also contain some managerial implications. First, it shows that advertising investment can be increased by introducing contracts, that is, to strengthen the partnership between manufacturers and shared manufacturing platforms to improve market capacity and overall competitiveness. So that participants can obtain more benefits with the same advertising investment. Second, the choices of advertising methods and the income obtained by manufacturers and shared manufacturing platforms are affected by the pricing of

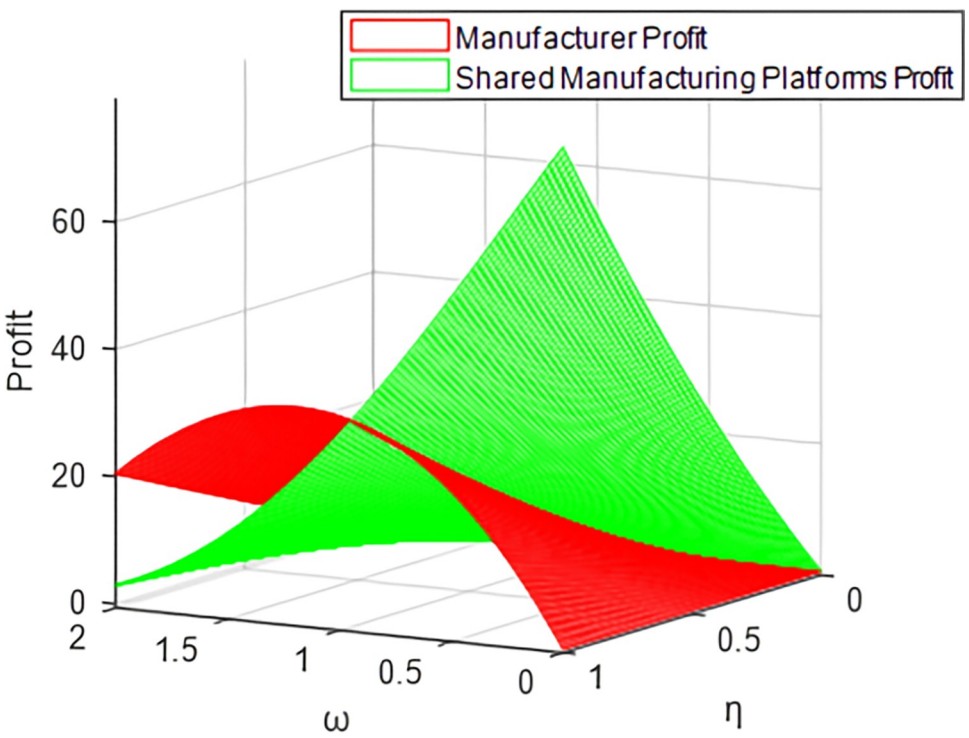

**Fig 7. Profits of manufacturers and platforms under the traditional cooperation model.**

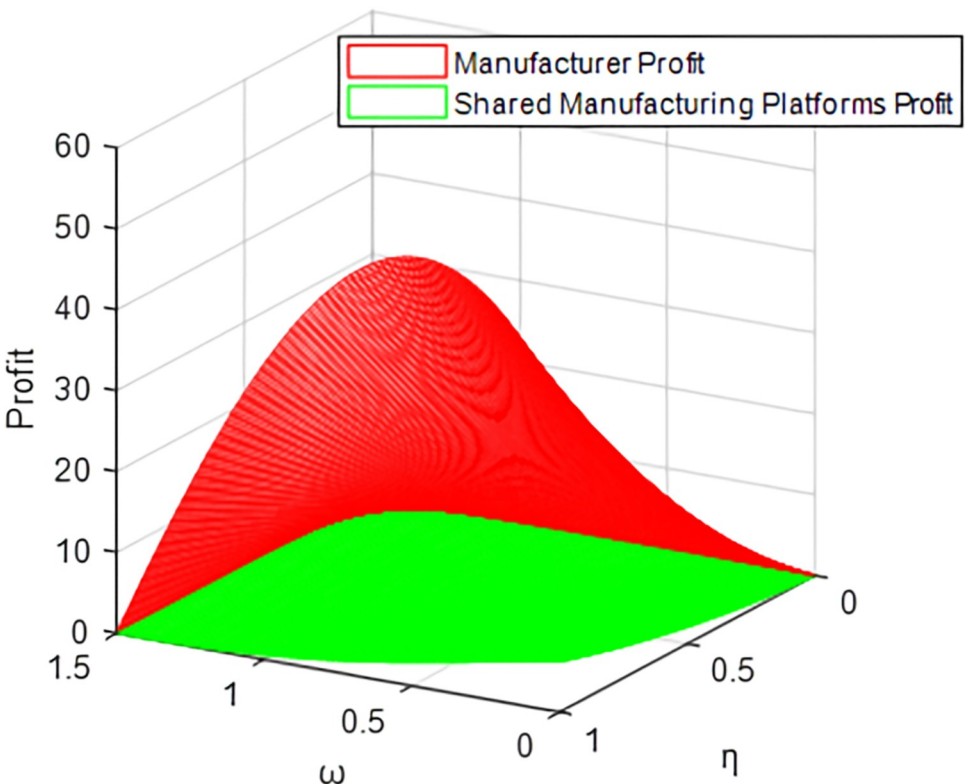

**Fig 8. Profits of manufacturers and platforms under the revenue sharing model.**

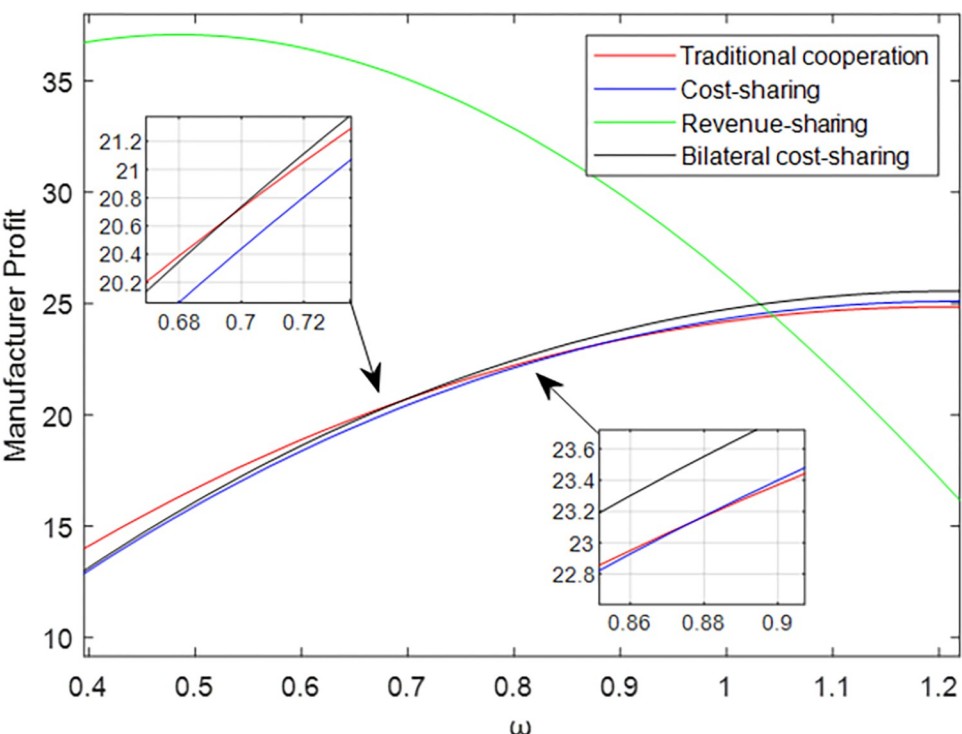

**Fig 9. Profit of manufacturers pricing $\omega$ under different models.**

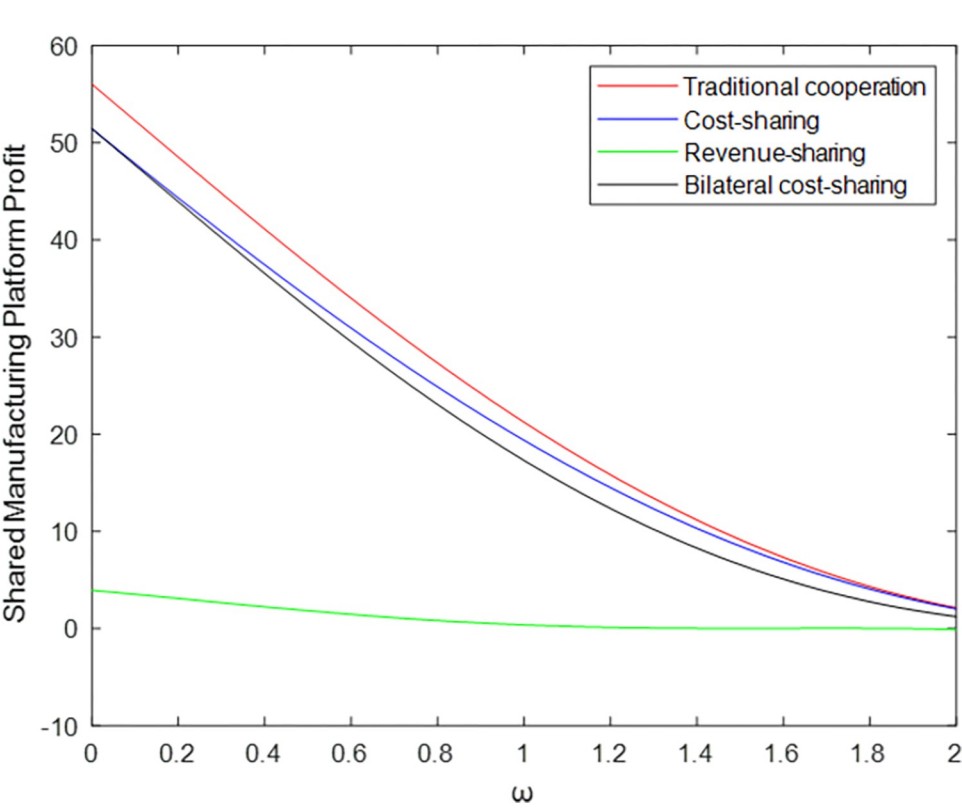

**Fig 10. Profit of shared manufacturing platforms under different models.**

manufacturers, and the government can regulate the shared manufacturing benefits by subsidizing manufacturers or increasing taxes. Third, there will be different behavioral decisions under the different pricing of manufacturers and shared manufacturing platforms, that is, there will be a shield when the contract is concluded. It is necessary to coordinate both parties to ensure overall coordination. So that the members of shared manufacturing can get more benefits in the context of the continuous development of shared manufacturing.

In conclusion, this paper analyzes the optimal pricing strategy of the shared manufacturing platform under different contract modes based on the cooperative advertising theory, and provides a theoretical reference for the shared manufacturing platform to formulate a reasonable pricing strategy. The impact of platform pricing and the market demand of manufacturers is also considered when establishing the pricing model, which is in line with the actual situation of the platform operation, which makes the pricing model in this paper more realistic.

The future work of this research is as follows. Firstly, the case of multiple manufacturers cooperating with a shared manufacturing platform at the same time will be discussed. Secondly, it is to analyze the possible competition between the two parties' advertising investments.

## Supporting information

**S1 Data.**
(DOCX)

## Author Contributions

**Formal analysis:** Peng Liu.

**Investigation:** Yantong Wu.

**Methodology:** Peng Liu.

**Writing – original draft:** Yantong Wu.

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
