## [Decision Letter · Decision Letter 0]

12 Nov 2023

PONE-D-23-19325Pricing Strategies for Shared Manufacturing Platform Considering Cooperative Advertising Based on Differential GamePLOS ONE

Dear Dr. Liu,

Thank you for submitting your manuscript to PLOS ONE. After careful consideration, we feel that it has merit but does not fully meet PLOS ONE’s publication criteria as it currently stands. Therefore, we invite you to submit a revised version of the manuscript that addresses the points raised during the review process.

We look forward to receiving your revised manuscript.

Kind regards,

Vijay Kumar

Academic Editor

PLOS ONE

Journal Requirements:

"This work was supported by the Key Program of Social Science Planning Foundation of Liaoning Province under Grant L21AGL017. The authors wish to acknowledge the contribution of Liaoning Key Lab of Equipment Manufacturing Engineering Management, Liaoning Research Base of Equipment Manufacturing Development, Liaoning Key Research Base of Humanities and Social Sciences: Research Center of Micro-management Theory of SUT."

"Acknowledgements: This work was supported by the Key Program of Social Science Planning Foundation of Liaoning Province under Grant L21AGL017. The authors wish to acknowledge the contribution of Liaoning Key Lab of Equipment Manufacturing Engineering Management, Liaoning Research Base of Equipment Manufacturing Development, Liaoning Key Research Base of Humanities and Social Sciences: Research Center of Micro-management Theory of SUT."

Funding information should not appear in the Acknowledgments section or other areas of your manuscript. We will only publish funding information present in the Funding Statement section of the online submission form. 

"This work was supported by the Key Program of Social Science Planning Foundation of Liaoning Province under Grant L21AGL017. The authors wish to acknowledge the contribution of Liaoning Key Lab of Equipment Manufacturing Engineering Management, Liaoning Research Base of Equipment Manufacturing Development, Liaoning Key Research Base of Humanities and Social Sciences: Research Center of Micro-management Theory of SUT."

**Additional Editor Comments:**

Kindly see the reviewers' comments below in this email.

Reviewers' comments:

Reviewer's Responses to Questions

**Comments to the Author**

1. Is the manuscript technically sound, and do the data support the conclusions?

Reviewer #1: Partly

Reviewer #2: Partly

2. Has the statistical analysis been performed appropriately and rigorously? 

Reviewer #1: N/A

Reviewer #2: N/A

3. Have the authors made all data underlying the findings in their manuscript fully available?

Reviewer #1: Yes

Reviewer #2: Yes

4. Is the manuscript presented in an intelligible fashion and written in standard English?

Reviewer #1: Yes

Reviewer #2: Yes

5. Review Comments to the Author

Reviewer #1: Summary:

This paper proposes four related hierarchical models of how companies can share manufacturing facilities, and analyze what the properties of these models imply about long-term strategic benefits in each case. Each model is quadratic and solvable in closed-form, and conclusions are based upon numerical results.

Strengths:

Overall, I find the topic of this paper to be interesting and a worthy contribution to the literature in this field. I am particularly interested in how the results of this paper may guide corporate and regulatory strategy in the area of shared manufacturing.

Major weaknesses:

- The models proposed are essentially unfounded and unjustified, apart from one reference which remains unexplained. For these results to be of serious interest, the manuscript must go further to explain “why is this model reasonable? Why should a practitioner believe in this model? When does the model break?”

- Experimental results are difficult to interpret. It would help to clearly state the following in each subsection of the numerical results: (a) what is the relationship we should expect from the interaction we are trying to model, (b) how do we construct an experiment to verify that relationship, (c) what is the result, and (d) interpret why the result does or does not support the intuition/expectation. Currently, some subsections state some of these points, but it is inconsistent.

Other weaknesses:

- The abstract and introduction could do a much better job of strongly motivating the topic, orienting the reader to the problem, etc. It is very difficult to parse, as is.

- The related work discussion is also not doing as good a job as it could: currently, it reads as a list of (seemingly arbitrary) papers with at best a loose organization. This discussion needs to orient the reader first to the general trends of thought in this field, and only then to the (relatively few) specific, closely-related works upon which the present paper is directly based. This structure may be present but it is hidden in lists of references and summaries.

- Minor suggestion: use LaTeX for proper equation formatting

- It seems problematic to assume manufacture production cost is zero. Is this really a good assumption? When does it break?

- The sharing coefficients in Assumption 4 do not clearly correspond to any model that has been introduced yet. This makes them difficult to interpret.

- It is unclear what the authors mean precisely by “two stage Stackelberg game” above Thm. 1. What are the stages, and how do they map onto the continuous time axis? Is this not a standard LQ game for which a feedback Stackelberg equilibrium is sought? What is the information pattern of this game? Similar questions for Thm. 2 and others.

- In Corollary 3, I do not understand the use of the term “saddle point.” If this is describing the graph of a SISO function, it is unclear what function is being referred to. If referring to a game-theoretic equilibrium: are these not general-sum problems (making the term “saddle” inappropriate)? What does “stable” mean in this context?

Reviewer #2: A. Overview

This research explores the concept of shared manufacturing, a novel business form that integrates production and manufacturing. The study introduces a model comprising a manufacturer and a shared manufacturing platform, employing cooperative advertising through various models. Using a differential game approach, the research analyzes the influence of key parameters on prices and profits, demonstrating the model's viability through numerical examples. The findings offer valuable decision-making insights for manufacturers and shared manufacturing platforms operating under diverse cooperative advertising strategies.

B. Comments

I have several crucial observations concerning the quality of the manuscript. The following are my comments:

1. Introduction

a. The introduction section lacks clarity in its presentation, particularly in articulating the significance and novelty of the manuscript.

b. The first important gap in the introduction lies in the limited exploration of the specific dynamics and challenges associated with the intermediary platform among the shared manufacturing platforms. Please improve the introduction section with the addition of relevant business examples.

c. While the study acknowledges its significance, a deeper analysis of the intermediary platform's operational intricacies, unique characteristics, and potential obstacles is needed for a comprehensive understanding. The critical analysis of relevant cases can be added.

d. Another notable gap is the absence of a detailed examination of the effectiveness and impact of cooperative advertising within the context of shared manufacturing platforms, particularly focusing on the intermediary platform. Please add the detailed examination of literature and cases.

e. The comprehensive summary of literature review is missing. Please add the table for the summary of literature review.

f. Please highlight the research gaps and mention research questions, which are not clear in the current form of manuscript.

g. Kindly provide justification for the utilization of cost-sharing, revenue-sharing, and bilateral cost-sharing contracts in the manuscript, supported by relevant business examples and literature.

h. What justifies the application of differential game theory in this paper, and what specific gaps or limitations in existing methodologies or frameworks does the use of differential game theory address within the context of shared manufacturing platforms?

2. Problem descriptions and assumptions

a. What specific gaps in current research exist regarding the selection of cooperation models for advertising investment in shared manufacturing scenarios, considering the leasing of idle resources and the involvement of both the manufacturer and shared manufacturing platform? It is not clear to me, please justify with the relevant business examples.

b. Why is Assumption 1 made, positing that under the cooperative advertising model, the manufacturer and the shared manufacturing platform have distinct advertising goals with a collaborative effect on market expansion and product sales improvement? Are there specific real-world scenarios or settings that validate or necessitate such an assumption in the context of shared manufacturing platforms.

c. Please articulate Assumption 4 clearly, providing a concise explanation supported by transparent cost-sharing equations.

3. Model analysis and solution

a. Clarify the process of deriving all equilibrium results and ensure that detailed proofs for all theorems are included. Currently, only the proof for Theorem 1 is found in the main body of the manuscript. Consider presenting important results in the main body and relocating all proofs to the appendix or supplementary section.

b. The detailed proofs for all corollaries are absent. It is recommended to present proofs in the appendix or supplementary section, with comprehensive explanations included in the main body of the manuscript, following the convention observed in high-quality journals.

c. Given Corollary 1, which asserts that as market development capabilities increase, advertising efforts by the manufacturer and shared manufacturing platform also increase, and that higher manufacturer pricing leads to an initially increasing and then decreasing trend in advertising efforts and resulting profits, what real-world cases or empirical evidence can be cited to justify these observed trends?

d. How do these dynamics align with practical scenarios, and can existing business cases provide insights into the implications of market, and profits in shared manufacturing platforms? Please justify all theorems and corollaries.

4. Comparative analysis

a. Please explain the Comparative analysis in more detail citing relevant business examples.

b. In the Table 4, increasing, decreasing, and no change representation is not clear, for example, p^(T*) is increasing with increase in α or decrease in α. Please make the table more comprehensive by adding nature of change of α,β,ω,and φ.

5. Numerical Analysis

a. Please add more examples for setting the value of relevant parameters. Are all parameters in line with first order and second order conditions?

b. Please enhance the clarity and quality of the figures, particularly the line representing bilateral cost-sharing, which is currently unclear (in Figure 2). Consider adding legends to distinguish between multiple lines for improved interpretation.

6. Conclusion

a. The conclusion section requires substantial improvement, particularly in terms of the quality of presentation, with a focus on clearly articulating the significant contributions of the study on the pricing strategies for shared manufacturing platform.

b. Consider incorporating two subsections in the manuscript: one to emphasize theoretical contributions and another to address practical/managerial implications. The current version lacks these essential components.

Overall, this manuscript is interesting but not publishable in its current form and needs major revision. Hope my comments are useful and important for the publication in top quality journal.

Thanks.

6. PLOS authors have the option to publish the peer review history of their article (what does this mean?). If published, this will include your full peer review and any attached files.

Reviewer #1: No

Reviewer #2: No

---

## [Author Response · Author response to Decision Letter 0]

26 Dec 2023

Dear Editor,

Thank you for allowing a major revision of our manuscript (PONE-D-23-19325) for publication in PLOS ONE., with an opportunity to address the reviewers’ comments.

We are uploading (a) our point-by-point response to the comments, (b) an updated manuscript with red highlighting indicating changes, and (c) a clean updated manuscript without highlights.

We thank the referee and the editor very much for the comments and suggestions. We have made considerable effort on revising the paper based on these comments. For future communications, please contact either P. Liu at the addresses listed below.

Yours sincerely,

Peng Liu

Professor Peng Liu 

School of Management

Shenyang University of Technology 

Shenyang 110870

China

E-mail: liupeng@sut.edu.cn

Response to the Comments of Reviewer #1 on PONE-D-23-19325

We thank the referee very much for the comments and suggestions. They are very helpful for us to revise and improve the paper. The paper has been carefully revised according to the referee’s advice. We have made the following changes on the paper accordingly.

Reviewer#1, Concern # 1: Overall, I find the topic of this paper to be interesting and a worthy contribution to the literature in this field. I am particularly interested in how the results of this paper may guide corporate and regulatory strategy in the area of shared manufacturing.

Author response: We thank the referee very much for the comment. 

From Corollary 3 and Figure 2, it can be seen that the revenue-sharing model maintains the highest market capacity in a stable state, followed by the bilateral subsidy model, the cost subsidy model, and finally the traditional cooperation model. Therefore, in order to maintain better advertising results, enterprises in the field of shared manufacturing should give priority to other cooperative advertising models in addition to traditional cooperative advertising; Figure 5 shows the benefits of enterprises in the field of shared manufacturing under different models, from which it can be seen that when the manufacturer's pricing is small, the manufacturer chooses the revenue sharing contract and the profit is higher, and when the manufacturer's price is higher, the bilateral cost subsidy is selected, and the profit is higher.

In general, in the context of the continuous development of shared manufacturing, only by continuously strengthening the source of shared resources and shared media, that is, the cooperative relationship between manufacturers and shared manufacturing platforms, and establishing a sound contract model (such as the bilateral cost-sharing contract and revenue sharing contract in the text), can participants get more benefits with the same advertising investment.

The managerial implications can be found on pages 29-30 in the revised paper.

Reviewer#1, Concern # 2: The models proposed are essentially unfounded and unjustified, apart from one reference which remains unexplained. For these results to be of serious interest, the manuscript must go further to explain “why is this model reasonable? Why should a practitioner believe in this model? When does the model break?”

Author response: We thank the referee very much for the comment. 

As suggested by the referee, we have added the descriptions of research significance in the section “2. Problem descriptions and assumptions”.

This paper considers the convexity of advertising costs, the Nerlove-Arrow model, the linear relationship between price factors and market demand, and completes the model establishment based on the references mentioned in this paper.

[1] Erickson G M. A differential game model of the marketing-operations interface. European Journal of Operational Research,2011,211(2):394-402.

[2] Zhang Jie, Chiang W K, Liang Liang. Strategic pricing with reference effects in a competitive supply chain. Omega,2014,44(2):126-135.

[3] Ouardighi F E. Supply quality management with optimal wholesale price and revenue sharing contracts: a two-stage game approach. International Journal of Production Economics, 2014,156(5):260-268.

[4] Chen G, Zhang X. Research on cooperative advertising strategy in dual-channel supply chain under brand competition. Technology Economics & Management Research, 2019, 5:8-13.

We updated the manuscript with the yellow highlighted changes on pages 8-9 in the revised paper.

Reviewer#1, Concern # 3: Experimental results are difficult to interpret. It would help to clearly state the following in each subsection of the numerical results: (a) what is the relationship we should expect from the interaction we are trying to model, (b) how do we construct an experiment to verify that relationship, (c) what is the result, and (d) interpret why the result does or does not support the intuition/expectation. Currently, some subsections state some of these points, but it is inconsistent.

Author response: We thank the referee very much for the comment. 

As suggested by the reviewers, we added an expectation of numerical results in Section 5, and the results successfully validated the validity of the model established in this paper.

 We updated the manuscript with the yellow highlighted changes on page 25 in the revised paper.

Reviewer#1, Concern # 4: The abstract and introduction could do a much better job of strongly motivating the topic, orienting the reader to the problem, etc. It is very difficult to parse, as is.

Author response: We thank the referee very much for the comment. 

As suggested by the reviewer, we have added a description of the importance of cooperative advertising in the introduction.

 We updated the manuscript with the yellow highlighted changes on page 1 and page 2 in the revised paper.

Reviewer#1, Concern # 5: The related work discussion is also not doing as good a job as it could: currently, it reads as a list of (seemingly arbitrary) papers with at best a loose organization. This discussion needs to orient the reader first to the general trends of thought in this field, and only then to the (relatively few) specific, closely-related works upon which the present paper is directly based. This structure may be present but it is hidden in lists of references and summaries.

Author response: We thank the referee very much for the comment. 

As suggested by the referee, we have improved the descriptions of "the current state of literature research" in the introduction.

We updated the manuscript with the yellow highlighted changes on pages 1-3 in the revised paper.

Reviewer#1, Concern # 6: Minor suggestion: use LaTeX for proper equation formatting

Author response: We thank the referee very much for the comment. We have improved proper equation formatting.

We updated the manuscript with the yellow highlighted changes on pages 13 -18 in the revised paper.

Reviewer#1, Concern # 7: It seems problematic to assume manufacture production cost is zero. Is this really a good assumption? When does it break?

Author response: We thank the referee very much for the comment. 

The purpose of this paper is to compare and analyze the decision-making of different models and provide a decision-making basis for shared manufacturing enterprises by establishing models of different cooperation modes and obtaining model equilibrium solutions. Manufacturing costs are not within the range of variables considered in this article, and are equivalent to constants in the calculations and have no effect on the comparison results. This approach to the assumption that the manufacturing cost is zero can also be seen in the following literatures.

[1] Chen G, Zhang X. Research on cooperative advertising strategy in dual-channel supply chain under brand competition. Technology Economics & Management Research, 2019, 5:8-13.

[2] Huo L, Wu J. Research on Competitive advertising strategy of double channel supply chain under influence of goodwill. Computer Engineering and Applications, 2020, 56(07):260-265.

Reviewer#1, Concern # 8: The sharing coefficients in Assumption 4 do not clearly correspond to any model that has been introduced yet. This makes them difficult to interpret.

Author response: We thank the referee very much for the comment. 

As suggested by the referee, we have improved the descriptions of "Assumption 4 " in the section “2. Problem descriptions and assumptions”. The cost-sharing coefficients of Assumption 4 correspond to the three models of 3.1, 3.2 and 3.4 in this paper, respectively.

 We updated the manuscript with the yellow highlighted changes on pages 9-10 in the revised paper.

Reviewer#1, Concern # 9: It is unclear what the authors mean precisely by “two stage Stackelberg game” above Thm. 1. What are the stages, and how do they map onto the continuous time axis? Is this not a standard LQ game for which a feedback Stackelberg equilibrium is sought? What is the information pattern of this game? Similar questions for Thm. 2 and others.

Author response: We thank the referee very much for the comment. 

As suggested by the referee, we have improved the descriptions of the solution process in the model analysis and solution. The purpose of the model is to determine the optimal level of advertising investment for both the manufacturer and the shared manufacturing platform to maximize their respective profits.

We updated the manuscript with the yellow highlighted changes on page 11 and page 15 in the revised paper.

Reviewer#1, Concern # 10: In Corollary 3, I do not understand the use of the term “saddle point.” If this is describing the graph of a SISO function, it is unclear what function is being referred to. If referring to a game-theoretic equilibrium: are these not general-sum problems (making the term “saddle” inappropriate)? What does “stable” mean in this context?

Author response: We thank the referee very much for the comment. 

As suggested by the referee, we corrected the word " a stable point " in Corollary 3. The point of the article is that W(t) stabilizes at a fixed value over time.

We updated the manuscript with the yellow highlighted changes on page 22 in the revised paper.

Response to the Comments of Reviewer #2 on PONE-D-23-19325

We thank the referee very much for the comments and suggestions. They are very helpful for us to revise and improve the paper. The paper has been carefully revised according to the referee’s advice. We have made the following changes on the paper accordingly.

Reviewer#2, Concern # 1: 1. Introduction

a. The introduction section lacks clarity in its presentation, particularly in articulating the significance and novelty of the manuscript.

b. The first important gap in the introduction lies in the limited exploration of the specific dynamics and challenges associated with the intermediary platform among the shared manufacturing platforms. Please improve the introduction section with the addition of relevant business examples.

c. While the study acknowledges its significance, a deeper analysis of the intermediary platform's operational intricacies, unique characteristics, and potential obstacles is needed for a comprehensive understanding. The critical analysis of relevant cases can be added.

d. Another notable gap is the absence of a detailed examination of the effectiveness and impact of cooperative advertising within the context of shared manufacturing platforms, particularly focusing on the intermediary platform. Please add the detailed examination of literature and cases.

e. The comprehensive summary of literature review is missing. Please add the table for the summary of literature review.

f. Please highlight the research gaps and mention research questions, which are not clear in the current form of manuscript.

g. Kindly provide justification for the utilization of cost-sharing, revenue-sharing, and bilateral cost-sharing contracts in the manuscript, supported by relevant business examples and literature.

h. What justifies the application of differential game theory in this paper, and what specific gaps or limitations in existing methodologies or frameworks does the use of differential game theory address within the context of shared manufacturing platforms?

Author response: We thank the referee very much for the comment. As suggested by the referee, we have revised the descriptions of Introduction. 

a. As suggested by the referee, we have improved the descriptions of " novelty of the manuscript " in the introduction. We updated the manuscript on page 6 in the revised paper.

“In summary, compared with the existing literature on shared manufacturing, the contributions of this paper are: (1) considering the long-term and dynamic characteristics of the impact of advertising investment on product demand, using differential games to study the pricing and advertising strategies under shared manufacturing from the perspective of long-term dynamics; (2) introducing a variety of contracts under the premise of cooperative advertising, and comparing and analyzing the decision-making of shared manufacturing enterprises under different contracts and the impact of different coefficients on the results.”

b. As suggested by the referee, we have improved the descriptions of " the intermediary platform among the shared manufacturing " in the introduction. We updated the manuscript on page 2 in the revised paper.

“This type of intermediary shared manufacturing platform (as shown in Figure 1) does not own its own manufacturing resources and capabilities, such as workers, equipment, and materials. Manufacturing service providers transform idle manufacturing capacity to connect idle manufacturing capacity to the platform to form a virtual "cloud factory". The manufacturing buyer publishes the production order on the platform, and the supply and demand sides finally reach a deal through independent search and negotiation on the platform. The advantages of the intermediary sharing platform are that the platform is more flexible and the user entry threshold is low; The disadvantage is that Internet companies have a lack of understanding of the cross-border manufacturing industry. Typical application cases include "Haizhi Online", which is a leading non-standard parts manufacturing and sharing platform in China, which is mainly responsible for the docking of domestic small and medium-sized parts processing enterprises and global procurement resources. The buyer uploads the parts drawings and order information, calculates the corresponding average market price through the "intelligent price checker", and if the buyer accepts the price, it will match the appropriate factory for it, and finally the factory will make a specific offer. This model helps to shorten the procurement cycle and improve the production efficiency of the factory. In addition, there are Tao factories and Zhibu interconnection.”

c. As suggested by the referee, we have improved the descriptions of " a deeper analysis of the intermediary platforms " in the introduction. We updated the manuscript on page 2 in the revised paper.

d. As suggested by the referee, we have improved the descriptions of " a detailed examination " in the introduction. We updated the manuscript on page 5 in the revised paper.

e. As suggested by the referee, we have added the comprehensive summary of literature review. We updated the manuscript on pages 4-6 in the revised paper.

 “From the above literature, it can be seen that the difference between the discussion of pricing in the existing literature is that different state variables are considered, and different platform pricing models are established by changing different charging models or profit subjects. The essence of these models is to analyze the equilibrium solutions of the models under different cost sharing methods or benefit distribution methods, so

---

## [Decision Letter · Decision Letter 1]

26 Feb 2024

PONE-D-23-19325R1Pricing Strategies for Shared Manufacturing Platform Considering Cooperative Advertising Based on Differential GamePLOS ONE

Dear Dr. Liu,

Thank you for submitting your manuscript to PLOS ONE. After careful consideration, we feel that it has merit but does not fully meet PLOS ONE’s publication criteria as it currently stands. Therefore, we invite you to submit a revised version of the manuscript that addresses the points raised during the review process.

We look forward to receiving your revised manuscript.

Kind regards,

Tinggui Chen

Academic Editor

PLOS ONE

Additional Editor Comments:

Thank you for submitting your manuscript to PLOS One.

I have completed my evaluation of your manuscript. The reviewers recommend reconsideration of your manuscript following major revision. I invite you to resubmit your manuscript after addressing the comments below.

Reviewers' comments:

Reviewer's Responses to Questions

**Comments to the Author**

1. If the authors have adequately addressed your comments raised in a previous round of review and you feel that this manuscript is now acceptable for publication, you may indicate that here to bypass the “Comments to the Author” section, enter your conflict of interest statement in the “Confidential to Editor” section, and submit your "Accept" recommendation.

Reviewer #1: All comments have been addressed

Reviewer #2: All comments have been addressed

Reviewer #3: (No Response)

2. Is the manuscript technically sound, and do the data support the conclusions?

Reviewer #1: Yes

Reviewer #2: Yes

Reviewer #3: Yes

3. Has the statistical analysis been performed appropriately and rigorously? 

Reviewer #1: N/A

Reviewer #2: N/A

Reviewer #3: Yes

4. Have the authors made all data underlying the findings in their manuscript fully available?

Reviewer #1: Yes

Reviewer #2: Yes

Reviewer #3: Yes

5. Is the manuscript presented in an intelligible fashion and written in standard English?

Reviewer #1: Yes

Reviewer #2: Yes

Reviewer #3: Yes

6. Review Comments to the Author

Reviewer #1: Thank you very much for addressing my comments, and particularly for the work you have done to clarify modeling assumptions.

Reviewer #2: Dear Authors,

I am perfectly satisfied with the quality of revision, and accept the manuscript.

With regards!

Reviewer #3: In a word, I find the topic of this paper to be interesting and a worthy contribution to the literature in this field. But there are still some problems that need further consideration.

1.Abstract does not clearly reflect the innovation and importance of the research, it can be summarized better.

2.The introduction should be summarized better and stimulate the topic better.

3.Figure 1 includes the retailer, why is the price p in equation 2 not the retailer pricing, and it's about Shared manufacturing platform pricing?

4.Why do the hypothesis 2 appear Dw and DG?

5.It seems to be something wrong with equation 4.

6.In Corollary 5, the authors mentioned that “With the increase of the manufacturer's price ω, the profit of the manufacturer increases first and then decreases”, Can you figure out the range of ω?

7.The ordinates of Figure 3(a), Figure 3(b), and Figure 3(c) are not clear.

8.More management implications can be added to the conclusion.

7. PLOS authors have the option to publish the peer review history of their article (what does this mean?). If published, this will include your full peer review and any attached files.

Reviewer #1: No

Reviewer #2: No

Reviewer #3: No

---

## [Author Response · Author response to Decision Letter 1]

31 Mar 2024

Dear Editor,

Thank you for allowing a major revision of our manuscript (PONE-D-23-19325R1) for publication in PLOS ONE., with an opportunity to address the reviewers’ comments.

We are uploading (a) our point-by-point response to the comments, (b) an updated manuscript with red highlighting indicating changes, and (c) a clean updated manuscript without highlights.

We thank the referee and the editor very much for the comments and suggestions. We have made considerable effort on revising the paper based on these comments. For future communications, please contact either P. Liu at the addresses listed below.

Yours sincerely,

Peng Liu

Professor Peng Liu 

School of Management

Shenyang University of Technology 

Shenyang 110870

China

E-mail: liupeng@sut.edu.cn

Response to the Comments of Reviewer #1 on PONE-D-23-19325R1

We thank the referee very much for the comments and suggestions. They are very helpful for us to revise and improve the paper. The paper has been carefully revised according to the referee’s advice. We have made the following changes on the paper accordingly.

Reviewer#1, Concern # 1: Thank you very much for addressing my comments, and particularly for the work you have done to clarify modeling assumptions.

Author response: We thank the referee very much for the comment. 

Response to the Comments of Reviewer #2 on PONE-D-23-19325R1

We thank the referee very much for the comments and suggestions. They are very helpful for us to revise and improve the paper. The paper has been carefully revised according to the referee’s advice. We have made the following changes on the paper accordingly.

Reviewer#2, Concern # 1: I am perfectly satisfied with the quality of revision, and accept the manuscript.

Author response: We thank the referee very much for the comment. 

Response to the Comments of Reviewer #3 on PONE-D-23-19325R1

We thank the referee very much for the comments and suggestions. They are very helpful for us to revise and improve the paper. The paper has been carefully revised according to the referee’s advice. We have made the following changes on the paper accordingly.

Reviewer#3, Concern # 1: Abstract does not clearly reflect the innovation and importance of the research, it can be summarized better.

Author response: We thank the referee very much for the comment. 

As suggested by the reviewer, we have added a description of the innovation and importance of the research in the abstract.

We updated the manuscript with the yellow highlighted changes on page 1 in the revised paper.

Reviewer#3, Concern # 2: The introduction should be summarized better and stimulate the topic better.

Author response: We thank the referee very much for the comment. 

As suggested by the reviewer, we have added a description in the introduction. 

We updated the manuscript with the yellow highlighted changes on page 6-7 in the revised paper.

Reviewer#3, Concern # 3: Figure 1 includes the retailer, why is the price p in equation 2 not the retailer pricing, and it's about Shared manufacturing platform pricing?

Author response: We thank the referee very much for the comment. 

Figure 1 illustrates the entire operating model of an intermediary shared manufacturing platform, including manufacturers, shared manufacturing platforms, and retailers. In this paper, a secondary model consisting of a shared manufacturing platform is established. We use this model to analyze their interests and cooperation. And the retailers' demand is influenced by the price of a shared manufacturing platform. So the price p in equation 2 is not the retailer pricing, and it's about shared manufacturing platform pricing.

Reviewer#3, Concern # 4: Why do the hypothesis 2 appear Dw and DG?

Author response: We thank the referee very much for the comment. We have changed“ ”to“ ” in the revised paper.

 We updated the manuscript with the yellow highlighted changes on page 9 in the revised paper.

Reviewer#3, Concern # 5: It seems to be something wrong with equation 4.

Author response: We thank the referee very much for the comment. 

 In the revised paper, we put Equation 4 in Assumption 6. The coefficients, and , are explained in Assumption 5. Compared to the original paper, we adjusted the order of the assumptions 5 and 6 in the revised paper. And there is no problem with Equation 4 after the adjustment.

We updated the manuscript with the yellow highlighted changes on pages 9-10 in the revised paper.

Reviewer#3, Concern # 6: In Corollary 5, the authors mentioned that “With the increase of the manufacturer's price ω, the profit of the manufacturer increases first and then decreases”, Can you figure out the range of ω?

Author response: We thank the referee very much for the comment. 

As suggested by the reviewer, we have added the range of ω in Corollary 5. 

We updated the manuscript with the yellow highlighted changes on page 24 in the revised paper.

Reviewer#3, Concern # 7: The ordinates of Figure 3(a), Figure 3(b), and Figure 3(c) are not clear.

Author response: We thank the referee very much for the comment. 

We updated the manuscript with the yellow highlighted changes on pages 26-27 in the revised paper.

Reviewer#3, Concern # 8: More management implications can be added to the conclusion.

Author response: We thank the referee very much for the comment. 

We added management implications in the conclusion. 

We updated the manuscript with the yellow highlighted changes on pages 30-31 in the revised paper.

---

## [Decision Letter · Decision Letter 2]

3 May 2024

Pricing Strategies for Shared Manufacturing Platform Considering Cooperative Advertising Based on Differential Game

PONE-D-23-19325R2

Dear Dr. Liu,

We’re pleased to inform you that your manuscript has been judged scientifically suitable for publication and will be formally accepted for publication once it meets all outstanding technical requirements.

Kind regards,

Tinggui Chen

Academic Editor

PLOS ONE

Additional Editor Comments (optional):

Reviewers' comments:

Reviewer's Responses to Questions

**Comments to the Author**

1. If the authors have adequately addressed your comments raised in a previous round of review and you feel that this manuscript is now acceptable for publication, you may indicate that here to bypass the “Comments to the Author” section, enter your conflict of interest statement in the “Confidential to Editor” section, and submit your "Accept" recommendation.

Reviewer #3: All comments have been addressed

Reviewer #4: All comments have been addressed

Reviewer #5: (No Response)

Reviewer #6: All comments have been addressed

2. Is the manuscript technically sound, and do the data support the conclusions?

Reviewer #3: Yes

Reviewer #4: Yes

Reviewer #5: Yes

Reviewer #6: Yes

3. Has the statistical analysis been performed appropriately and rigorously? 

Reviewer #3: Yes

Reviewer #4: Yes

Reviewer #5: Yes

Reviewer #6: Yes

4. Have the authors made all data underlying the findings in their manuscript fully available?

Reviewer #3: Yes

Reviewer #4: Yes

Reviewer #5: Yes

Reviewer #6: Yes

5. Is the manuscript presented in an intelligible fashion and written in standard English?

Reviewer #3: Yes

Reviewer #4: Yes

Reviewer #5: Yes

Reviewer #6: Yes

6. Review Comments to the Author

Reviewer #3: The author is very careful about the revisions. I am satisfied with the author's response and intend to accept the manuscript.

Reviewer #4: The authors have revised the manuscript according to the reviewer's opinion, which basically meets the requirements. There are no other comments for now. The revised version is in a better shape for publication at PLOS ONE. Thus, I accept the paper for PLOS ONE.

Reviewer #5: The research objectives and problem statement of the thesis are clear.

The research method is clear, and there are sufficient data and conclusions to prove it.

The structure of the thesis is reasonable and logical.

Reviewer #6: Authors modified manuscript significantly. Graphical explanation is improved. Hence I recommend this work for publication.

7. PLOS authors have the option to publish the peer review history of their article (what does this mean?). If published, this will include your full peer review and any attached files.

Reviewer #3: No

Reviewer #4: No

Reviewer #5: No

Reviewer #6: No

---

## [Editor Report · Acceptance letter]

10 May 2024

PONE-D-23-19325R2 

PLOS ONE

Dear Dr. Liu, 

I'm pleased to inform you that your manuscript has been deemed suitable for publication in PLOS ONE. Congratulations! Your manuscript is now being handed over to our production team.

Kind regards, 

on behalf of

Dr. Tinggui Chen 

Academic Editor

PLOS ONE